# Shape evolution of pumice during granular flow
Carolina Figueiredo [1] ✉, Ulrich Kueppers [1], Luiz Pereira [1,2], Lisa Depauli[1], Sarp Esenyel[1] & Donald B. Dingwell [1,2]

Explosive volcanic eruptions are a major geo-hazard. Given the energetic nature of eruptive processes, direct observation is limited, making the study of deposits and pyroclast textures essential for understanding eruption dynamics. Experimental constraints therefore provide a vital contribution to improving hazard assessment. We performed tumbling experiments using pumice lapilli from the Laacher See eruption (Eifel, Germany) to investigate ash generation and pyroclast shape evolution. Before and after each experimental step, samples were sieved, and the volume and four morphological parameters (axial ratio, convexity, form factor, solidity) of 100 clasts were measured. Most shape change happened before the first 15 min (first experimental step) and produced up to 48 wt.% ash. We frame our analysis in terms of effective relaxation timescales, whereby pyroclasts display a decelerating rate of shape change towards a time-invariant morphology. This quantification of the susceptibility of porous pyroclasts to changes enhances our understanding of transport processes from clast generation to sedimentation.

During the explosive eruption of felsic magma, fragmentation is typically dominated by brittle failure[1,2]. The efficiency of this process depends on textural characteristics (e.g., porosity, crystallinity, permeability), magma ascent conditions, and the transient gas overpressure in pore space resulting from the competition between degassing and outgassing[3]. Upon fragmentation, angular porous pyroclasts of variable grain size are accelerated and eventually ejected into the atmosphere[4,5]. The eruptive and transport dynamics control the spatial distribution of the pyroclastic deposits. Fall (gravitational settling dominated) and flow (sub-horizontal transport) generates the two end member deposit types of such explosive events. The meaning of the grain size distribution as revealed from volcanic deposits is a matter of discussion as it may depend—at variable degrees—on primary (magma fragmentation) and post-fragmentation transport processes. Pyroclast shape may be altered during conduit flow, turbulence in the eruption plume, impact during sedimentation, particle-particle and particle-substrate interaction during pyroclastic density currents (PDCs) and deposit compaction.

Fall deposits present characteristics that are related to the effects of transport-related sorting due to atmospheric conditions (e.g., wind field, humidity) as well as particle density and aerodynamic properties[6–9]. Such deposits contain predominantly angular clasts and the sedimentological characteristics exhibit evidence of density- and/or grain size-sorting. As a consequence of this process, in medial to distal regions, these deposits are commonly well-sorted. In contrast, PDCs travel down volcanic flanks, experiencing both density-stratification and turbulence. Particle concentration and flow turbulence define a wide spectrum of these currents, ranging from particle-poor (i.e., pyroclastic surges) to granular flows, transport regimes with high granular temperature. In the latter, particle-particle and particle-substrate interactions are anticipated to be ubiquitous[10–13]. Porous pyroclasts accordingly undergo evolution in size and shape[14–16] while becoming smaller and rounded. The volcanic ash content, either part of a flow from the beginning or formed in-situ, affects the flow's mobility, enlarging the areas at threat.

The efficiency and impact of the experimental settings on such size and shape evolution of pyroclasts have been investigated experimentally via (1) tumbling[17–20] and (2) drop[21,22] experiments as well as with the aid of numerical model[15]. Clast abrasion and morphology evolution have also been evaluated for gravel particles in dry and wet settings, relevant to debris flows[23,24] and bed load transport in rivers[25]. These studies indicate a strong influence of experimental conditions on clast shape evolution and the resulting dynamic coupling to the surrounding medium[26]. In this sense, in principle, experiments are capable of accessing the variation of morphological parameters. Independent of the experimental conditions and material properties, a widely recognised principle of particle abrasion remains: If a particle does not fracture during a process, then its tendency to change shape decreases over time eventually reaching a time-invariant 'equilibrium state' in terms of shape and roughness[27].'Relaxation' is describing the decreasing velocity of the change of a parameter.

[1]Department of Earth and Environmental Sciences, Ludwig-Maximilians-Universität München, Munich, Germany. [2]GEOLAB, Hangzhou International Innovation Institute, Beihang University, Hangzhou, China. ✉e-mail: figueiredo.ca@lmu.de

The physical concept of relaxation was initially proposed to describe the dynamics of gases and applied to dielectric materials. Nowadays, it has been used across various systems, including glass science, volcanic processes, superconductors, jamming in granular materials as well as biological and magnetic systems[1,28–32]. It is vastly used to study structural relaxation of glasses, whereby a structure and consequently the material properties evolve to equilibrium at a rate dictated by a characteristic relaxation time[33,34]. During relaxation, the rate of change of a property commonly decelerates as this property approaches the equilibrium or optimal state[33]. Here, we investigate the shape relaxation of abrading lapilli using laboratory experiments.

## Results

We have conducted tumbling experiments under controlled conditions (Section 4.1) and quantified the related abrasion process in terms of the ash fraction and a series of morphological parameters of particle shape (Section 4.2). We have analysed the observed fine particle generation (Section 2.1) and clast shape change (Section 2.2) in terms of relaxation times for shape and roughness evolution.

### Ash generation

We present the results of dry tumbling experiments (Fig. 1) conducted at room temperature and ambient pressure, expanding an empirical correlation presented by Hornby et al.[18]. We used pumice lapilli (2 <x < 64 mm) from Laacher See (East Eifel, Germany; sourced from pristine fall deposits of the 13 ka phonolitic Plinian Laacher See eruption near Nickenich, Germany). Stratigraphically, they belong to the Lower (LLST) and Middle (MLST) Laacher See Tephra units and contain 10–15 vol% (dense rock equivalent) phenocrysts (sanidine and accessory mafic minerals)[35].

A strong non-linear relationship between tumbling duration (displayed as tumbling distance—calculated using drum diameter and experiment duration[18]) and ash generated was observed for three experimental conditions of the so-called *Set A* (T1, T2, T3B) and one for *Set B* (T1) experiments. In T1, ash is returned to the drum after each time step. In T2, ash is removed after each time step. While in T3B, ash is also removed after each time step, but steel balls are added to the drum. The produced ash was quantified as the weight fraction of experimentally generated volcanic ash (based on the starting weight) and is displayed cumulatively in Fig. 1.

Ash generation was highest at the beginning of the experiments (within the first 1000 meters of tumbling) and decreased non-linearly with increasing experimental duration (cf. Figure 1). This trend can be visualised as a "relaxation" of ash generation efficiency, dependent on the bulk contact energy per experiment type (see Section 4). A sense of the magnitude of these changes in our experiments is provided by the observation that amongst all experiments, T3B produced the largest amount of ash (up to

48 wt%), followed by T2 (up to 36 wt%), and T1 (up to 25 wt%). These results document the prime influence of bulk contact energy on ash generation. The presence of inter-clast ash led to reduced ash generation (T1 compared to T2) while the steel balls locally increased point loads that lead to abrasion (dominant) or break-up (T3 compared to T2).

### Morphological analysis

We have evaluated one physical (volume) and four morphological (*axial ratio* [particle elongation], *convexity* [fine-scale textural roughness], *form factor* [overall particle irregularity] and *solidity* [morphological roughness]) parameters that describe the shape of 100 clasts at each experimental step and its evolution with time and experiment type. We have cast the analysis of these experimental results in terms of relaxation theory, yielding a novel insight into particle abrasion. Key parameters that govern the evolution of volcanic pyroclastic particle shapes are generated, providing a useful way for volcanologists to understand transport and sedimentation processes (and thereby eruption dynamics) based on thorough analysis of deposits.

More than 2,400 measurements were performed to quantify the shape and physical properties of pumice lapilli. The morphological evolution of clast shape became clear when quantifying shape parameters via 2D image analysis: *axial ratio*, *convexity*, *form factor*, and *solidity*. In *Set A*, we analysed 100 randomly selected clasts after each time step of tumbling experiments (i.e., 0' [starting material SM], 15', 30', 45', 60', and 120') (Fig. 2). From the mean values of *convexity*, *form factor* and *solidity* (Box plots, Fig. 2A, 2B and 2C), we observe most shape evolution in the first 15 min of tumbling.

*Set A* proved that shape evolution is very rapid in the tumbling device. Regarding overall ash generation (Fig. 1), we observe a relaxation of shape evolution as a function of bulk contact energy per experiment type (Fig. 2). To constrain the observed changes more in detail and rule out any influence of clast heterogeneity when randomly picking 100 clasts, we designed a *Set B* series of experiments (Fig. 3) (for type T1 experiment) and dyed 100 lapilli clasts with food colour, thereby enabling their individual identification and sampling after each experimental increment. Additionally, the shorter time intervals (5', 10', 15', 20', 60') between analytical steps enabled an enhanced resolution of shape evolution with time. Volume decreased steadily (Fig. 3A), whereas *axial ratio* remained constant, confirming the uniform abrasion of tumbling (Fig. 3B). The higher time resolution for *Set B* experiments over *Set A* is well illustrated by the evolution of the *convexity* (Fig. 3C), *form factor* (Fig. 3D) and *solidity* (Fig. 3E) parameters. The most substantial changes occurred in the first 5 min. After 20 and 60 min of tumbling, a very limited number of the starting 100 lapilli had broken as indicated by the increased number (106 and 109, respectively) of coloured lapilli that could be identified. As the percentage of particles that underwent breakage was relatively small, these breakage events were not frequent enough to generate abrupt parameter changes in the particle population

**Fig. 1 | Ash production during tumbling experiments.** Plotted as cumulative ash fraction (weight fraction of the starting mass) against the rotational distance for experiments T1, T2 and T3B (Set A) and T1 (Set B). Result reproducibility is being shown by 2 data sets each (this study) and in accordance with earlier studies[18]. Set B experiment T1 with coloured clasts reproduces the ash generation trend, proofing the subordinate impact of the food colouring on the mechanical properties of the 100 coloured clasts.

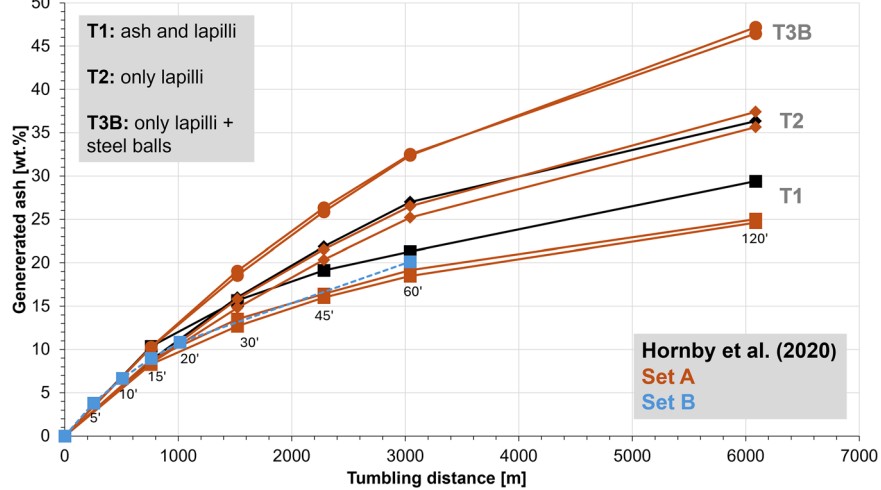

**Fig. 2 | Morphological changes in pyroclasts during Set A experiments.** Particle evolution is shown separately for SM (starting material), T1, T2, and T3B as boxplots for **A** *convexity*, **B** *form factor* and **C** *solidity*. The median is represented by the horizontal line in the box, while the mean is denoted by a cross. Whiskers extend to 1.5 times the interquartile range, with data points beyond this range considered potential outliers. The inclined parallel lines on the x-axis indicate a break in the time series and a change in sampling distribution.

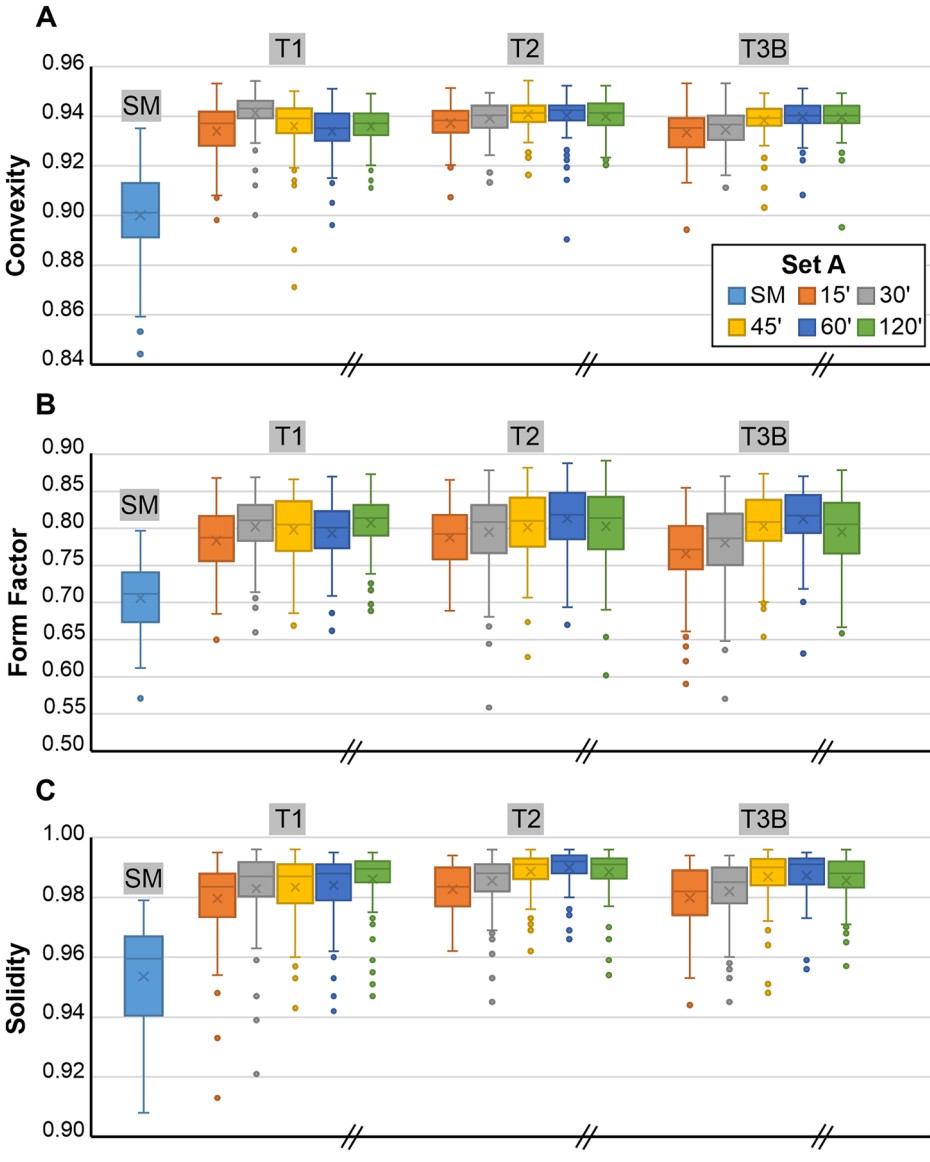

parameters (Fig. 3). The shape evolution throughout the experiments is also evident with kernel density estimations (KDE) when plotting *form factor* vs. *axial ratio* for *Set A* (Fig. 4A) and *Set B* (Fig. 4B), and is further illustrated with the progressive smoothing of contours in clasts shadowgraphs. However, the overall end shape values do not differ substantially between the experiments. The importance of tracking the same batch of particles using the techniques employed here (*Set B*) is apparent in our experimental results (Figs. 2 and 3) since it will allow for modelling particle abrasion.

## Discussion
### Tumbling experiments
Our experimental set-up is related but not identical to the techniques of other studies in volcanology[17,19,20] and clastic sediment transport[24,25] and therefore can be seen as an innovative approach for experimental volcanology assessments of granular flow. For experiments with a comparable horizontal rotational axis, continuous mixing without clast segregation had been observed[36]. Accordingly, for the experimental geometry used here (with slightly inclined rotational axis and two bed-inverting "ladles"), it is assumed that particles were in dominantly rolling (i.e., subordinate saltation or sliding) motion even though clasts are varying in size (i.e., 2–5 cm length) and porosity (ca. 45–90%). Ash generation was observed in all experiments and at all experimental durations.

Particle shape changes and the related generation of fine particles are expected during any kind of transport regime where clasts interact with one another and/or free surfaces (e.g., substrate, wall) at sufficient contact energy. In volcanic currents, abrasion can be expected to be ubiquitous with evidence having been recorded from both comparatively dense clasts in block-and-ash flows or lahars[23] and mechanically weak porous lapilli[20,37]. The starting conditions (e.g., initial mass, initial temperature, grainsize distribution, peak velocity) affect the mobility of these currents in which gravitational settling is offset by turbulence. Substantial amounts of ash particles delay gas/air escape from the current and maintain mobility. Ash is constantly depleted from a current by deposition as well as elutriation to the overriding ash cloud. However, there is a continuous (yet variable) replenishment of ash generated due to particle-particle and particle wall/substrate-interactions that also affects transport behaviour[38].

Hornby et al.[18] produced a preliminary investigation of ash production and morphological evolution in pyroclastic density flows or granular flows. Here, we have built on that study in several ways. We have, most importantly, (1) substantially increased the experimental parameter space, (2) quantified both ash generation and pyroclast shape evolution and (3) added a further lithology (angular fall lapilli clasts from Laacher See compared to "Waschbims" (industrially wet sieved)[18]). These experiments serve to map quantitatively the impact of flow conditions on ash generation efficiency and

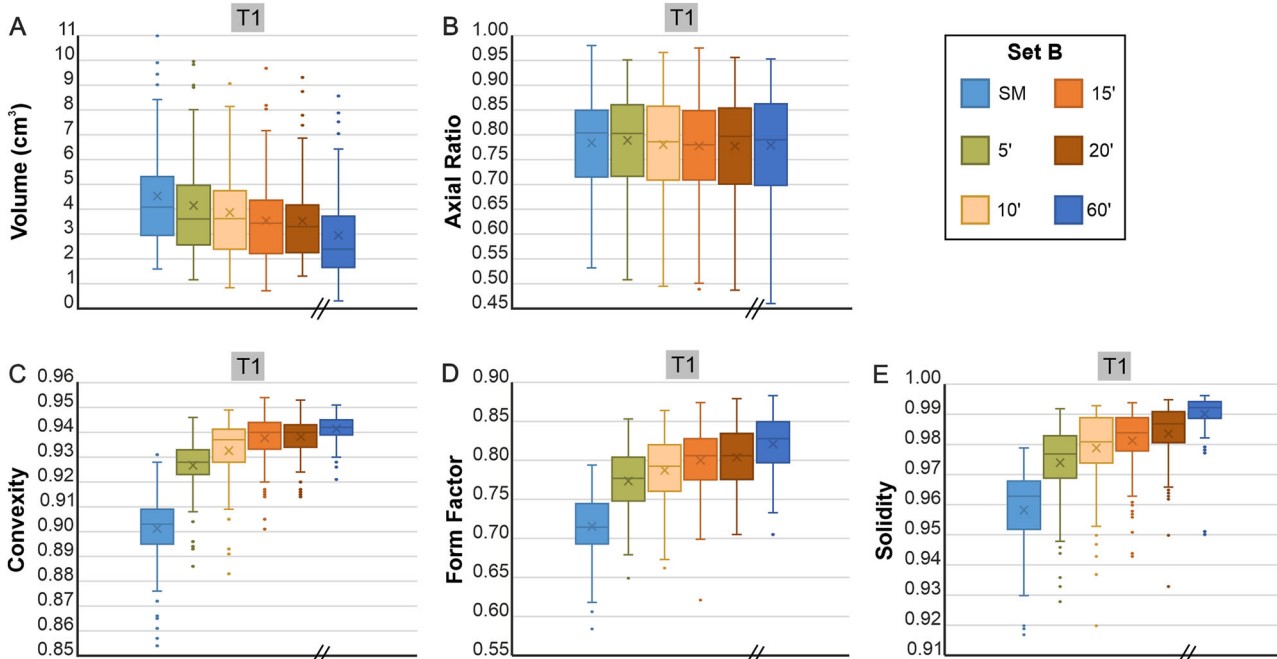

**Fig. 3 | Morphological changes in pyroclasts during Set B (containing 100 coloured particles) tumbling experiments.** Boxplot for evolution of **A** volume, **B** *axial ratio*, **C** *convexity*, **D** *form factor* and **E** *solidity* are included. Volume has been calculated from the three measured axes. The median is represented by a horizontal line, while the mean is denoted by a cross. Whiskers extend to 1.5 times the interquartile range, with data points beyond this range considered potential outliers and depicted as coloured dots. SM: Starting material. The inclined parallel lines on the x-axis indicate a break in the time series and a change in sampling distribution.

the related change of shape parameters of the transported (i.e., tumbled) pumice lapilli under controlled and repeatable conditions. These experiments are conducted at constant conditions and do therefore not account for the dynamic variations in nature such as changing transport velocity, gas flux, temperature, particle concentration and/or grain-size variability. Laboratory experiments often require designs simpler than nature in order to enable the unequivocal derivation of empirical laws and the individual influence of single parameters. As expected, experiments with periodic ash removal, corresponding in the natural case to elutriation to the ash cloud overriding a PDC or sedimentation, cumulatively produced more ash (T2 and T3B), reflecting the fact that the accumulated ash absorbs collision energy, reduces the intensity of clast interactions and therefore particle abrasion. Adding two stainless-steel balls (220 g each, representing approx. 10% of the starting cargo) in experiment T3B, increased ash production, regardless of ash removal. This underlines the contribution of dense clasts (frequently considered to be lithics) in promoting the generation of fines during PDC transport. The initial stages of clast interaction are crucial for ash production, with mechanical energy and collision efficiency playing key roles.

Ash generation efficiency was found to be remarkably self-similar during the first 15 min, but subsequently quite distinct for the T1, T2, and T3B experiments. During the first 15 min, experimental conditions were identical for T1 and T2 experiments. The initial generation of ash and change of shape of angular pumice lapilli upon tumbling by overprinting edges and ridges and removing surface roughness asperities dominates sufficiently that the effects of the addition of steel balls (T3B) does not manifest itself in the data. Morphological analysis of *n* = 100 pyroclasts in *Set A* reveals that the parameters of *convexity, form factor,* and *solidity* exhibit a rapid evolution within the first 15 min of tumbling and only minor changes thereafter. This establishment of an apparent plateau was most pronounced in the T2 and T3B experiments, where ash was removed at each time step (Fig. 2).

The experience gained from the *Set A* experiments led us to conclude that a higher temporal resolution was required to effectively parameterise

shape evolution. This was addressed by designing *Set B* experiments, investigating ash generation and clast shape after 5, 10, 15, 20, and 60 min, respectively. Additionally, 100 random clasts of the starting cargo had been dyed red with food colour. This way, the same 100 clasts could be sampled after each experimental increment to capture detailed trends in volume reduction, visualised as changes of *convexity, form factor,* *axial ratio* and *solidity* (Fig. 3). The refined temporal trends confirm the strongly nonlinear shape evolution dynamics of pyroclasts. When plots of *axial ratio* against *form factor*, it becomes clear that all data sets exhibit similar features that are independent of the shape of the starting material. In detail however, the particle morphology is dependent on several experimental variables (i.e., number of particles, rotational speed, type of particles). While shape parameters in *Set A* experiments derive from randomly selected 100 clasts (Fig. 4A), the red colour allowed for easy sampling. Beside a better constrain on shape evolution (Fig. 4B), it became apparent that the number of coloured starting clasts (*n* = 100) increased to 106 (i.e., six lapilli broke apart) and 109 (i.e., nine lapilli broke apart) clasts after 20 and 60 min, respectively. The reason for this break-up is unclear. At this stage, it cannot be ruled out some clasts were pre-fractured. The breakage of these clasts was not significant enough to cause discontinuities or abrupt changes in the trends (Fig. 3), indicating that particle abrasion is the dominant mechanism driving shape evolution during the tumbling experiments.

## Significance of clast shape

During magmatic fragmentation, magmatic volatiles at overpressure in bubbles expand according to the highest pressure differential and eventually force the rising and deforming magma (viscous behaviour) into (local) brittle response to the acting stress. The liberated gas volume will continue expanding (in most cases towards Earth surface) and the related aerodynamic drag will accelerate the surrounding pyroclasts. After transport in the conduit, the gas-pyroclast jet will enter the atmosphere from the volcanic vent. Laboratory experiments mimicking explosive eruptions generate gas-only or gas-pyroclast jets whose characteristics depend on the starting conditions. Results include that (1) particle exit velocity strongly decays with

**Fig. 4 | *Axial ratio* vs. *form factor* plots for tumbling experiments.** Set A **A** and Set B experiments **B**. The dotted black line indicates the limit between starting material (SM) and tumbled clasts. The Kernel density estimation (KDE) to the right and the top of both plots distinctly allow for tracing the temporal shape evolution of the Set B clasts. Shadowgraphs of the specific clasts (**B**) show the decrease of surface roughness comparing starting material and tumbled clasts.

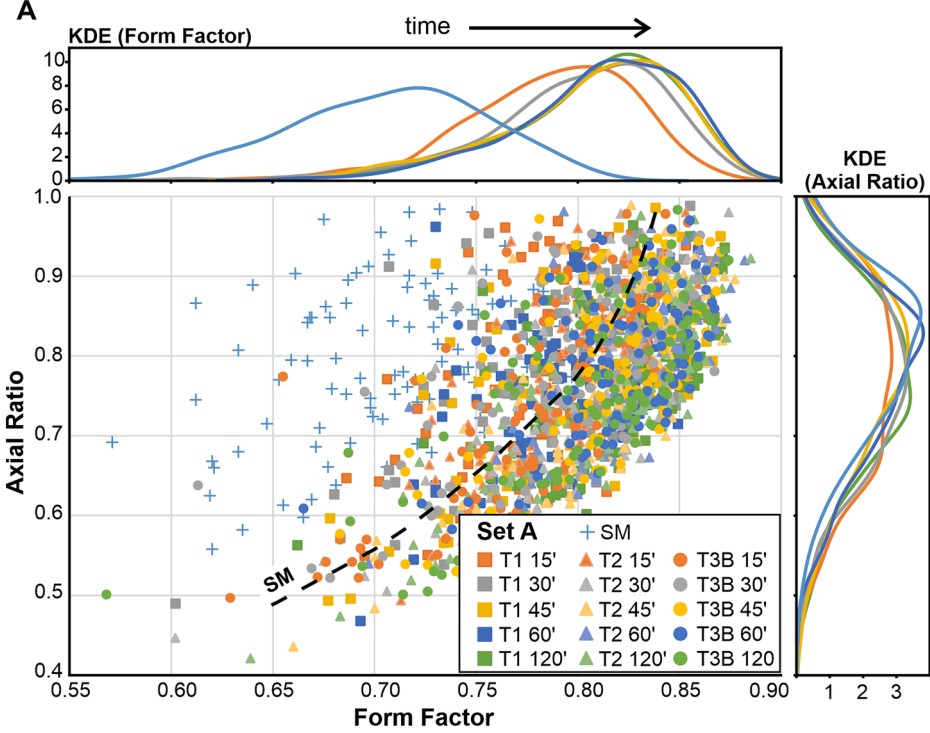

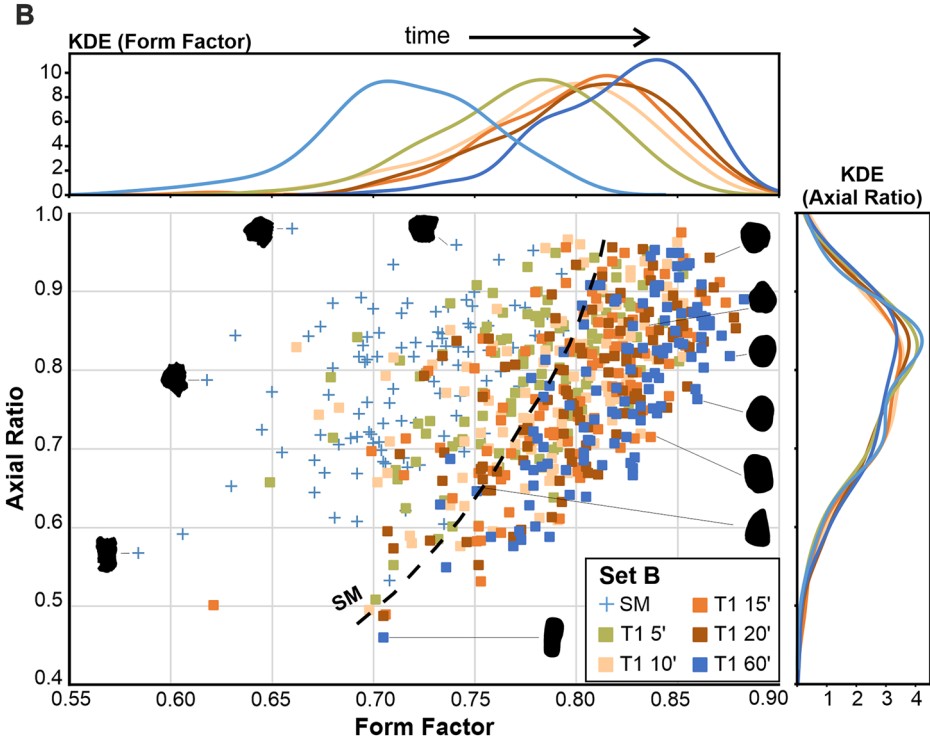

time[39] (this does not refer to particle deceleration) and (2) particles of any grain size travel at comparable velocities (up to 250 m/s with velocity differences rarely exceeding few m/s) when leaving the experimental vent[4,39]. We assume a similar dilatational flow field inside volcanic conduits from above the fragmentation level all the way to the surface. If this holds true, pyroclast-pyroclast collisions are unlikely or of low impact energy as clasts travel in the same direction. Hence, a statistically relevant shape description of lapilli-sized clasts would allow for describing if these clasts were exposed to post-fragmentation processes that can alter the primary shape.

Granular materials exhibit variability in both their morphological properties (e.g. grain shapes, size distributions, and surface characteristics) and their microstructural arrangements, including the organisation of grains and their contact points[40]. In our tumbling experiments, we do not address the microstructural arrangements due to the complexity and variability of how grains organise and interact at a small scale. Interactions between particles of dry, rough, and granular materials have been experimentally defined as being mainly due to friction and collision[41], and fragmentation and abrasion are the main processes that govern disruption of the

frictional materials[42]. In collisional interactions with a significant overlap of clast cross-sectional area ( > 50%), particle break-up is promoted, whereas in the tangential case (overlap <10%), impact energy is predominantly translated into rotational movement, yielding abrasion.

During most explosive volcanic eruptions, porous magma (per definition containing a melt fraction) is fragmented in a brittle way, generating angular fragments of variable size. Depending on clast temperature and melt composition, little or no (substantial) shape alteration driven by surface tension occurs after such fragmentation event. Fall deposits of such explosive eruptions are commonly dominated by angular clasts. We now compare our experimental findings to natural deposits and try to constrain which type of deposit may have been subject to processes that overprinted the angular shape of pyroclasts formed by brittle processes during magma fragmentation. This includes in general (1) in-conduit gas-particle flow, (2) kinetic and/or buoyant transport in eruption plumes, (3) (partially) collapsing columns, (4) PDCs, (5) sedimentation, and (6) deposit compaction[43]. Based on the propensity of porous pyroclasts to shape change observed in this study, we conclude that angular lapilli clasts as observed in fall deposits of explosive eruptions have not been subject to substantial processes capable of shape change. The low energy clast interactions in the experiments caused rapid shape change. At any process mentioned above (1–6) that could lead to clast shape change, collisions favour break up while tangential interactions would promote abrasion. We stress that we do not exclude particle-particle, particle-wall (conduit) or particle-substrate (PDCs) interactions to happen. Yet, with collision/tangential interaction processes potentially happening at the same temporal probability, disruption of two rounded clasts following collision would generate clasts with several fracture planes and at least one 'smooth' surface whose surface roughness had been altered. Accordingly, we believe that grain shape (and accordingly size) changes are of minor importance during conduit flow and eruption plume ascent as clasts with surfaces of two distinct characteristics are essentially absent in fall deposits. In PDCs, particle-particle and particle-substrate interactions are likely higher, leading to a higher probability of shape alteration. Especially in the denser underflow of PDCs, clast break-up and clast abrasion[38] will take place at ratios depending on PDC flow properties (e.g. particle volume fraction, flow velocity variations, breaks in slope angle, presence/concentration of lithic clasts). Rounded pumices within PDC deposits have been observed by the authors and are reported for e.g. Lascar Volcano (Chile)[44,45] and Novarupta (USA)[46]. Our experimentally derived data clearly demonstrate the high potential for porous pyroclasts to quickly (within minutes) change their shape. The laboratory experiments presented here are less energetic than those extant during volcanic currents (e.g., grain flows, PDCs), yet particle shapes have adjusted very rapidly, supporting the idea that shape change of pumice lapilli and the related ash production occur quite readily[20]. Careful field investigations of PDC deposits revealed that massive to stratified to cross-stratified facies occur, possibly ranging vertically and spatially at close range. This led to the model that the shear velocity of PDCs at the interface to the substrate ranges from substantial (leading to "passing by" [no deposition] or entrainment of clasts from the substrate and/or vegetation possible) to absent (fully depositional without noticeable erosive interaction with the aggrading deposit)[47]. Lapilli clasts from the same stratigraphic horizon have revealed substantial variations in shape characteristics. The fast shape evolution of porous lapilli clasts suggests that clasts of similar textural properties and different shape characteristics did not follow the same transport path until sedimentation.

**Relaxation timescale of shape evolution**

The effect of achieving an equilibrium of the smoothness of a surface is documented by the established plateaus of *convexity*, *form factor*, and *solidity* (Figs. 2 and 3), regardless of the experiment type. This phenomenon has been termed "equilibrium roughness" in the field of tribology[27]. It depicts how surfaces adapt during initial contact under sliding conditions to reach a state of minimum wear and friction.

Abrasion efficiency is interpreted as the rate of roughness change and, neglecting minor contributions from fragmentation, the sample can be treated as an approximation of a fluid continuum with roughness as an internal variable directly affected by viscosity, which can be defined as a measure of a fluid's resistance to flow[42,48–53]. Turbulent fluctuations create an energy cascade process, where the stress power from the mean flow dissipates across characteristic length scales of the flow. Literature contains several examples of complex equations controlled by several material properties used to model granular materials[40]. However, there is a lack of information on how the shape of these materials changes over time and of the implications of this in the flow dynamics. In this work we introduce a simple equation with three fitting parameters (i.e., $\tau_k, \beta, p_\infty$) to explain the shape evolution of frictional granular materials.

We propose that the equilibrium roughness phenomenon can be described by a relaxation law[30,31]. Perhaps the best studied example of the application of relaxation theory in volcanology is its application to the liquid-to-glass transition[1,2]. It has also been applied to the topic of viscous shape relaxation of vesicles and its impact on the suspension rheology of vesicular magmas[54–56].

In the present context, as considering the flowing granular material as a fluid continuum, we can treat the shape parameters—*convexity*, *form factor*, and *solidity*—in terms of relaxation times within the flow. These parameters account for large-scale and small-scale roughness and form, respectively, each presenting, in principle, a specific relaxation time.

When a material undergoes a certain change, any feature of physical property at a given time $p(t)$ evolves with aging time from its initial value $p_0$ to its final end value $p_\infty$, such that a normalised relaxation function $\phi(t)$ is the following:

$$\phi(t) = \frac{p(t) - p_\infty}{p_0 - p_\infty} \tag{1}$$

We employ the Kohlrausch-Williams-Watts (KWW) equation (Eq. 2) that describes the temporal function decay of a given property $p$[30,31]:

$$\phi(t) = \exp\left[-\left(\frac{t}{\tau_k}\right)^\beta\right], \tag{2}$$

where $\tau_k$ is the characteristic relaxation time, and $\beta$ describes the breadth of the relaxation time distribution, serving as the non-exponentiality parameter. In case $\beta$ is unity, it corresponds to a simple case of exponential decay, representing a single relaxation time, i.e., the process happens all over the material at the same rate. For $0 < \beta < 1$, it indicates a stretched exponential decay, meaning a distribution of relaxation times, where $\beta$ close to unity represents a narrow distribution, while $\beta$ close to zero represents a broad range of relaxation times across the material. A combination of Eqs. 1 and 2 gives rise to:

$$p(t) = p_\infty + (p_0 - p_\infty) \exp\left[-\left(\frac{t}{\tau_k}\right)^\beta\right] \tag{3}$$

We used the mean values of *convexity*, *form factor*, and *solidity* for each time increment (0, 5, 10, 15, 20, and 60 min) of experiment T1 (*Set* B) as the studied property $p$ (Fig. 5). 'Time increment 0' refers to the starting material (SM). The relaxation timescale for *convexity* is 5.1 min (Fig. 5A), *form factor* 8.7 min (Fig. 5B) and *solidity* 12.5 min (Fig. 5C), implying that finer-scale adjustments occur on shorter timescales than changes to the overall shape and structural roughness of the particles.

By analysing Eq. 3, $\tau_k$ refers to the time required for particles relaxation 63.2% of its path to equilibrium. As mentioned, $\beta$ reflects the distribution of relaxation times, which in glass science might be linked to the varying structures on atomic to nanoscale levels[57–59], therefore these different regions undergo relaxation following different rates. By analogy with the current system, we suggest that the $\beta$ values (Fig. 5) illustrate how a group of particles (here $n = 100$) undergoes shape relaxation. Values close to 1 mean that

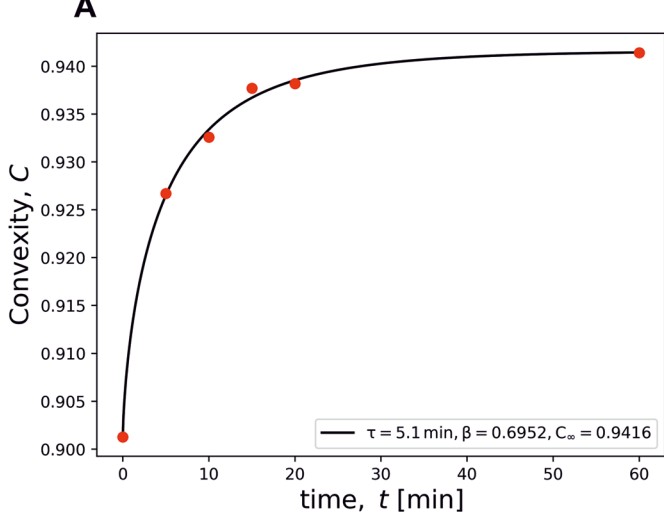

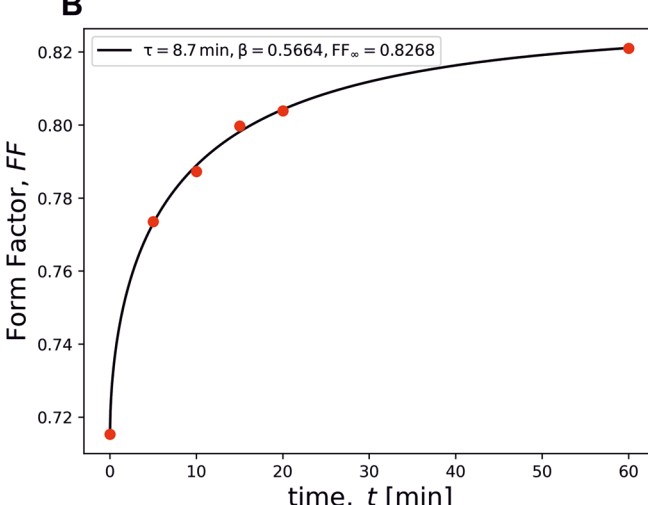

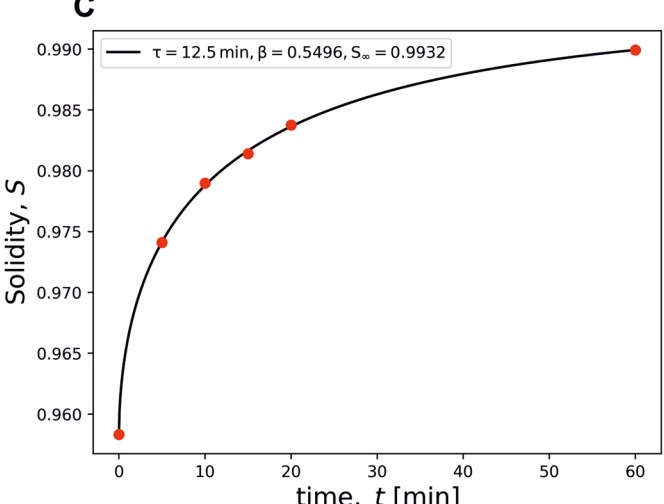

**Fig. 5 | Temporal evolution of particle shape parameters during tumbling experiment type T1.** The plots depict the variation of **A** *convexity*, C, **B** *form factor*, FF, and **C** *solidity*, S, over time t [min]. The solid black lines represent the best fit to the experimental data (red dots) using the relaxation model described by equation [1], while the initial (SM) and final values are provided within each plot.

particles change shape in a similar way (narrow distribution of shape relaxation time), while β approaching zero indicates a broad distribution of shape relaxation times. The analysis of this parameter can be a useful tool to evaluate how heterogeneous our set of particles is, characterising our system with β values « 1, not near 1, as different particles change shape in different rates (cf. Figure 2 and Fig. 3). These variations might be related to different textures of the starting materials, such as crystal and bubble content and initial shape.

It is interesting to note that the β and $\tau_k$ values for the three evaluated shape parameters (*C*, *FF*, and *S*) are not identical. This parallels the independent evolution of glass properties (e.g., density, refractive index)[60,61] during the relaxation process of silicate melts and glasses. In both systems, the evolution of any parameter is not a proxy for but rather a symptom of the relaxation process.

The parameterisation of the evolution of pyroclast morphology presented here improves our physical understanding of transport of pyroclastic materials, in detail the efficiency of particle-particle interaction and related ash generation. We believe this to be of great significance as collisional processes take place in a wide variety of volcanic settings. Pyroclasts formed by brittle fragmentation and preserved as angular fragments in fall deposits serve as proxy for natural processes and serve as reference point against with which pyroclasts collected from PDC deposits may be compared.

We suggest the use of relaxation theory to describe the shape evolution of particle in terms of *convexity, form factor* and *solidity*. Thorough quantitative analysis of clasts from primary volcanic deposits directly in the field will avoid any post-sampling size and shape alteration.

We recognise this study as an important step into further understanding pyroclast abrasion in granular flows. While our data provide valuable insights, future experiments with different starting materials (composition and texture), the study of natural deposits linked to the samples and the integration of other lithological properties (e.g. density, bubble number density) will aid with the applicability of the law.

We propose that this will allow for upscaling the empirical laws from laboratory experiments and contribute to enhanced understanding of transportation processes, in particular those affecting the mobility of pyroclastic density currents. Besides, we consider that this type of system can serve as an analogue model to investigate relaxation processes in more complex materials, such as glasses and supercooled liquids.

## Methods

Rotary tumbling experiments that mimic particle interactions during PDCs or granular flows have been used to study the abrasion of pumice lapilli. The particle interactions, albeit likely of lesser intensity than possible in natural transport, are caused by forced tumbling of the experimental charge. Previous studies[18–20] have empirically constrained the relationship between tumbling conditions and ash generation. To increase our mechanistic understanding of pyroclast size and shape evolution during PDC transport, we conducted further tumbling experiments as defined below. We extended the analysis of the experimental cargo and have evaluated a physical (volume) and four morphological (*axial ratio* [particle elongation], *convexity* [fine-scale textural roughness], *form factor* [overall particle irregularity] and *solidity* [morphological roughness]) parameters that describe the shape evolution during the experiments.

### Tumbling experiments and ash generation

The tumbling experiments were carried out at ambient conditions in an industrial cement mixer (see Hornby et al.[18] for detailed description). The starting material (SM) was sourced from fall deposits of the 13 ka phonolithic Plinian Laacher See eruption near Nickenich, Germany. Each experiment started with 2 kg of pumice lapilli with a porosity range of 45–93% -mean porosity 80%- and particle size of 1.12–5 cm. In *Set A*, three types of experiments were performed to map out abrasion efficiency as a function of collisional energy. The experiments were stopped after 15, 30, 45,

**Table 1 | Description of the experimental conditions of two sets and three types of tumbling experiments**

| Set | Experiment type | Description | Duration | Target clasts |
|---|---|---|---|---|
| A | T1 | Ash and lapilli returned to drum in each step for further tumbling | 15', 30', 45', 60', 120' | random (n = 100) |
| | T2 | Only lapilli returned to drum, ash stored separately in each step | | |
| | T3B | T2 + 2 steel balls (220 g each) | | |
| B | T1 | Ash and lapilli returned to drum in each step for further tumbling | 5', 10', 15', 20', 60' | coloured (n = 100) |

Collisional energy was varied by removing/keeping ash from/in the drum after each increment or adding steel balls. Clasts have been investigated after five time-steps (representing theoretical tumbling distance). In Set A, random clasts were selected for analysis, in Set B, the same batch of 100 clasts was investigated after each increment. Numbers in bold indicate time steps repeated in Set A and B.

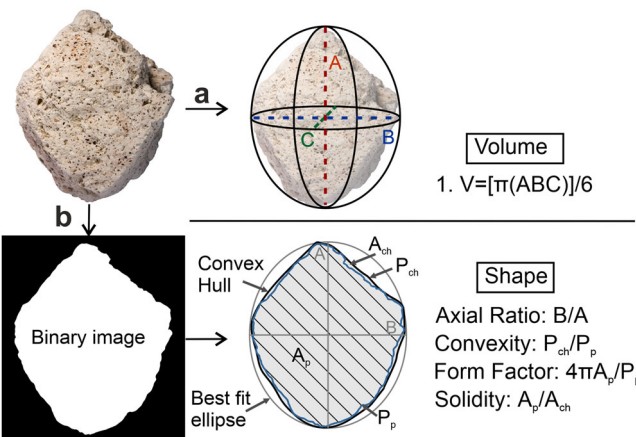

**Fig. 6 | Workflow to constrain the pumice clasts physical properties.** volume (**a**) and shape (**b**). **a** Calliper measurement of three orthogonal axes (A, B, and C) and volume calculation. **b** Illustration of the parametrization measurements (axial ratio, convexity, form factor and solidity) on a binary image of pumice clast.

60, and 120 min, respectively (Table 1). The entire sample was dry sieved and split at 2 mm. In T1 experiments, both size fractions of the sieving procedure described above (i.e. lapilli and ash generated) were put back into the drum after each time increment. In T2, only the remaining lapilli fraction was put back into the drum, the ash was sampled and stored. In T3B, we added two steel balls to mimic the frequent presence of higher-density (than pumice) clasts during natural transport. We chose to use steel balls (non-erodible) instead of natural rocks in order to be able to attribute 100% of generated ash to the abrasion of pumice.

For T1, we performed a second set of experiments (*Set B*), in which 100 lapilli from within the 2 kg starting material had been coloured with food colour prior to tumbling. This treatment did not affect the mechanical response as density and weight remained effectively unchanged. In this manner, shape analysis was performed on the same subset of the starting material. Ash generation was quantified by dry sieving at 2 mm after 5, 10, 15, 20 and 60 min, respectively.

**Morphological properties**

In addition to obtaining the weight fraction of ash generated per experimental increment, we also quantified clast volume (Fig. 6a), weight and morphology (Fig. 6b). Clast volumes were calculated using the manual method[62], measuring the length of three orthogonal axes with a calliper and calculating the volume assuming a three-axial ellipsoid geometry. For shape analysis, we photographed each clast independently on a light table. The resultant scaled shadowgraphs show the projected area of each clast. All images were processed on Photoshop (image inverting, clast contour delimitation and binarization) and analysed for their shape parameters with *ImageJ* (Fig. 6). We increased the resolution compared to earlier studies[17] by analysing macro images taken of each clast individually (and not in sets of n = 100). We selected *axial ratio, convexity, form factor* and *solidity*[63] to cover both roughness (small and large scale) and form of the particles and

used a adapted shape macro[64]. We chose to perform 2D analyses to obtain statistically robust, fast and low-cost data. For our analysis, we assume that the shape characteristics (not the 2D aspect ratio) are not orientation dependent. In *Set A*, 100 randomly selected clasts were taken from the lapilli fraction for analysis and afterwards put back to the lapilli cargo in the drum for the subsequent tumbling. In *Set B*, clasts coloured with food colour could be picked after each experimental increment for analysis and afterwards put back to the lapilli cargo in the drum for the subsequent tumbling.

While the results presented here were performed in the laboratory, volume and shape analysis will also be done on natural deposits directly in the field, thereby avoiding any effect of sample treatment during sampling, transport and storage. The method presented here (weighing, axes measurement, shadowgraph) is an easy and logistically feasible approach to produce statistically representative datasets in the field.

## Data availability

Source data for Figs. 1–5 of the manuscript are available in the Zenodo repository at https://doi.org/10.5281/zenodo.16737285 under a Creative Commons Attribution 4.0 International License (CC BY 4.0).

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

## Acknowledgements

CF and UK acknowledge support by grant KU2689/7-1 from the Deutsche Forschungsgemeinschaft. LP is grateful for the support of the Alexander von Humboldt Foundation. DBD, LP, & CF acknowledge the support of ERC 2018 ADV Grant 834225 (EAVESDROP) to DBD. We also thank the three anonymous reviewers and the editor for their constructive comments and suggestions, which greatly improved the manuscript. We are grateful to Prof. Dr. Kelly Russell for insightful discussions and feedback on the Set A data.

## Author contributions

CF and UK conceptualized the study. CF, UK, LD and SE performed the experiments and analysed the data. LP analysed the experimental data. CF and UK wrote the manuscript. DBD provided the funding. All authors discussed the findings, developed the analysis and generated the manuscript.

## Funding

## Competing interests

The authors declare no competing interests.
