## [Transparent Peer Review file · Communications Earth & Environment]

Shape evolution of pumice during granular flow

Corresponding Author: Ms Carolina Figueiredo

Version 0:

Decision Letter:

Dear Ms Figueiredo,

Your manuscript titled "Shape evolution of pumice during granular flow" has now been seen by 3 reviewers, and we include their comments at the end of this message. They find your work of interest, but some important points are raised. We are interested in the possibility of publishing your study in Communications Earth & Environment, but would like to consider your responses to these concerns and assess a revised manuscript before we make a final decision on publication.

In particular, please ensure that in the revised manuscript you fully explain your experimental configuration and its application to natural volcanic cases and that you justify why the grains were not analysed in 3D.

We therefore invite you to revise and resubmit your manuscript, along with a point-by-point response that takes into account the points raised. Please highlight all changes in the manuscript text file.

Please submit your point-by-point responses as a separate file, distinct from your cover letter where you can add responses to the Editors' comments that you do not want to be made available to the reviewers. Word files are preferred. We recommend that any figures, tables or graphs that are included in the response to reviewers are also included in the main article or Supplementary Information.

Please use the following link to submit your revised manuscript, point-by-point response to the referees' comments (which should be in a separate document to any cover letter), a tracked-changes version of the manuscript (as a PDF file) and the completed checklist:

Link Redacted

We hope to receive your revised paper within six weeks; please let us know if you aren't able to submit it within this time so that we can discuss how best to proceed. If we don't hear from you, and the revision process takes significantly longer, we may close your file. In this event, we will still be happy to reconsider your paper at a later date, as long as nothing similar has been accepted for publication at Communications Earth & Environment or published elsewhere in the meantime.

Please do not hesitate to contact us if you have any questions or would like to discuss these revisions further. We look forward to seeing the revised manuscript and thank you for the opportunity to review your work.

Best regards,

Domenico M. Doronzo, PhD
Editorial Board Member
Communications Earth & Environment

orcid.org/0000-0002-6866-8870

Joe Aslin
Deputy Editor
Communications Earth & Environment

EDITORIAL POLICIES AND FORMATTING

Editorial Policy: [Policy requirements](https://www.nature.com/documents/nr-editorial-policy-checklist.pdf) (Download the link to your computer as a PDF.)

- Behavioural and social science
- Ecological, evolutionary & environmental sciences
- Life sciences

<https://www.nature.com/documents/nr-reporting-summary.zip>

Furthermore, please align your manuscript with our format requirements, which are summarized on the following checklist: [Communications Earth & Environment formatting checklist](https://www.nature.com/documents/commsj-phys-style-formatting-checklist-article.pdf)

and also in our style and formatting guide [Communications Earth & Environment formatting guide](https://www.nature.com/documents/commsj-phys-style-formatting-guide-accept.pdf) .

*** DATA: Communications Earth & Environment endorses the principles of the Enabling FAIR data project (<http://www.copdess.org/enabling-fair-data-project/>). We ask authors to make the data that support their conclusions available in permanent, publically accessible data repositories. (Please contact the editor if you are unable to make your data available).

All Communications Earth & Environment manuscripts must include a section titled "Data Availability" at the end of the Methods section or main text (if no Methods). More information on this policy, is available at <http://www.nature.com/authors/policies/data/data-availability-statements-data-citations.pdf>.

If a community resource is unavailable, data can be submitted to generalist repositories such as [figshare](https://figshare.com/) or [Dryad Digital Repository](http://datadryad.org/). Please provide a unique identifier for the data (for example a DOI or a permanent URL) in the data availability statement, if possible. If the repository does not provide identifiers, we encourage authors to supply the search terms that will return the data. For data that have been obtained from publically available sources, please provide a URL and the specific data product name in the data availability statement. Data with a DOI should be further cited in the methods reference section.

REVIEWER COMMENTS:

Reviewer #1 (Remarks to the Author):

Dear editor,

In this contribution, the authors present a study of the evolution of shape and size of pumice clasts, as well as the generation of ash, during transport phases of granular flows, which is mainly based on tumbling experimental data. With lapilli clasts sampled from fallout deposits of the Laacher See eruption as the starting material, Figueiredo and colleagues designed and developed different sets of experiments aimed at understanding and quantifying the temporal evolution of a series of morphological parameters of the studied pyroclasts (e.g. axial ratio, convexity and others). The temporal evolution of these parameters during the experiments was described using an approach based on the definition of relaxation parameters. The manuscript is well-written, its structure allows for an ordered description of the presented information, methodology is clear and reproducible, and the manuscript integrates illustrative figures and tables that facilitate the description and interpretation of results. Although a few references are suggested in the commented PDF, the literature on this topic is properly recognized and well represented in the introduction and discussions of the manuscript. Said that, I would like to raise the following main points:

1. The authors show that, under their experimental conditions, shape of pyroclasts rapidly evolves due to abrasion towards a "final" state. In other words, the relaxation times tend to be short. From this, the authors conclude, among other observations, that abrasion in the conduit is probably small considering the irregular, angular shape shown by lapilli clasts in fallout deposits. I understand the logic behind this, but there are relevant processes, especially shortly after fragmentation, that may generate irregular morphologies such as clasts breaking and mask the morphological effects of abrasion due to the continuous generation of new irregular clasts. I think experiments may suggest that abrasion is small in the plume, but it is not clear to me this conclusion in the conduit. I suggest to discuss this topic.
2. How comparable are the experiments with granular flows that propagate with higher temperatures? This likely changes the mechanical properties of clasts, which are critical to define the efficiency of abrasion and clasts breaking.
3. The authors use steel balls to enhance abrasion (as done by lithics for instance) and, as expected, they have a relevant effect in experimental results. Again, it is not clear the natural conditions, if any, for which these experimental conditions are relevant.
4. I think Figure 2 can be modified in order to be clearer. Titles above SM are confusing. I suggest to present the labels "Convexity", "Form Factor" and "Solidity" in the y-axis, and a label of SM above the first box in each panel.
5. If the allowed number of figures limit allows it, I suggest to extend the methodology by including a figure with a scheme of the experimental device. On the other hand, the information of specific clasts in Figure 4 (I mean, the shadowgraphs) is too small and could be improved. Finally, in Figure 5, I think the variable tau should be expressed in time dimensions.

In summary, even if experiments are useful tools to evaluate the shape evolution dynamics of pyroclasts in granular flows, I think that results are maybe overinterpreted in terms of the analysis of natural volcanic phenomena because it is not clear under which conditions natural systems can be described by means of this experimental configuration. The authors mention that granular flows present physical characteristics for which we could expect an even higher abrasion/breaking capacity, but this should be discussed in detail. What is the solid fraction in experiments? Do we have an idea of energy involved in collisions? What kind of granular flow (or part of a granular flow) it is expected to represent?

All in all, I suggest publication of this manuscript after minor/moderate revisions, most of them related to the clear definition of the natural scenario the authors are simulating in the experiments, and eventually avoid overinterpretation of results.

There are several comments and small editorial suggestions in the attached PDF. Please note that my mother-tongue is not English.

Reviewer #2 (Remarks to the Author):

Dear Authors

I have reviewed the paper by Figueiredo et al., which outlines an experimental approach to defining the production of fines/ash-sized particles from pumice lapilli. The manuscript was extremely well written and logically presented, adhering to the conventions of the journal. I found no reason to make any grammatical adjustments to the paper, as any changes would have just been stylistic. The references provided seem adequate, but also highlight that the authors and other researchers have examined this process both in field studies and laboratory studies, and I would question the novelty of the work presented. However, the paper shows some interesting results, particularly the speed at which the ash is produced, <15-20 mins, which is surprising. It would have been beneficial to reproduce the results over many more experiments to confirm this result and look for similarities in field/real-world examples.

The major concern I have with this study and analysis is why weren't the grains mapped or characterised in 3D? The methods exist from simple 3D scanning techniques through to SEM, MCT, and even synchrotron analysis. The 2D characterisation of already fragmented and deposited grains will have a range of surface features/textures, porosity, and possibly crystal features that are not well represented in the measures undertaken, which may have some influence on the outcome. Surely, density, porosity, and some measure of hardness, glass surface structures would also be interesting to explore alongside grain surface textures/shapes than just analysing grain shape parameters. In my opinion, expanding the measures of the properties of the grains would advance this study.

The sample set of pumice lapilli is, in my opinion, too limited to be able to make broad conclusions about pyroclastic flow processes. I would expect that a range of pumice material was tested based on a range of pumice properties to see if the results are a standard response to the tumbling. I would expect that just using pumice from one unit within an eruption, where the material has already been fragmented and transported, may skew the results. While I understand that it is probably impossible to test newly fragmented, erupted lapilli before it has transformed into a flow, there must be some proximal pumice units worldwide that represent the erupted material of an eruption column that is coeval to a pyroclastic flow that would be a better target to sample.

As the experiment is meant to simulate ash generation in a pyroclastic flow, I would expect the discussion to have some more detail around where this process would occur in the flow (spatially and temporally) and how the fines contribute to the flowing mass and whether any sedimentological descriptions of real-world flow deposits show an increase in fines either laterally/longitudinally or vertically, however I understand the need to limit the amount of text in this format. I also wonder if the relationships found in Figure 5 can be related to the real-world flows/deposits or even experimental flows. This may be beyond the scope of this study, but I am sure if the research progresses, it would be an important comparison.

I am not sure of the relevance of further investigating the grain size of the newly produced ash and whether there needs to be further presentation and analysis of grain sizes below 2mm. In some flows, fines or clay-sized particles may contribute to changing flow conditions. I understand that moving beyond tumbling in a concrete mixer is most likely beyond this study, but characterising the volumes of grain sizes below 2 mm may inform future experimental analysis on flow behaviours.

I would also recommend removing any reference or text (lines ~250-290) that tries to relate these experiments to conduit processes or fragmentation, as I do not see how these experiments, tumbling of lapilli in a cement mixer, relate to or scale to the conduit fragmentation process. While I understand that ash is produced in those processes, the experimental design is probably more aligned to collision and (low energy) frictional processes in a flow. Focusing on pyroclastic flow processes also presents a more focused study and manuscript.

A minor change I could consider would be changing the heading at 4.2 to morphological properties, not petrophysical properties, as only shape properties have been determined; no lithological properties have been determined, such as porosity, etc.

The results and analysis of the manuscript support the conclusions. In general, the discussion could be more focused on pyroclastic flow processes, yet overall, the study does present some interesting starting points for future (postgraduate student) research. The manuscript will influence some further investigation into future thinking in the field and most likely influence the analysis of large-scale experimental flows and how they are analysed during flow, as well as deposits are analysed. The manuscript could be improved through recognising the limitations of the experiments, in particular the sample material gathered, and the lack of repeatability or the number of experiments conducted. Stronger justification of the clast shape parameters chosen could be provided, and why these are characteristics best related to the physical process of ash generation, and how other lithological parameters may provide a more holistic view of factors contributing to ash generation. Again, why the clasts were not analysed using a 3D method needs to be explained, and whether 3D analysis would present any further benefit would be helpful. The authors could also indicate how the results of these experiments could inform real-world deposit analysis.

Overall, the manuscript presents a well-constructed experiment with some results of interest to the wider field. The analytical approach and analysis presented are robust. I see no reason why the manuscript should not be published, as the results could inform future experimental approaches to understanding pyroclastic flow processes, which are currently not well understood.

I look forward to seeing the published manuscript.

Thank you

Reviewer #3 (Remarks to the Author):

The paper deals with the interpretation of experiments on abrasion of pumice clasts in a rock tumbler. The experiments analyze morphology characteristics and roughness of clasts, amount of ash produced, after variable tumbling times and in different conditions. The authors discuss many points, among which the most relevant are:

- lapilli abrasion is reduced by the presence of ash
- lapilli converge toward a stable morphology configuration

The authors discuss their results in terms of gas/magma /quenched pyroclast flow in the conduit and within PDCs.

The article is well organised and the results are thoroughly discussed. I had some difficulties following the results chapter as no detail was given on the experimental setting and protocols until the final methods chapter.

I believe the paper is almost ready to be published after some minor correction/changes.

Here are my recommendations:

- It is necessary to add a table with main experimental conditions (T1, T2, T3b) and results (for example, amount of ash produced) at the beginning of the results chapter to help readers follow data presentation.
- For better comparison, it is necessary that solidity, convexity and form factor axes in figures 2 and 3 display the same

intervals and size.

-morphology: Convexity, form factor and solidity do not significantly (as seen from boxplots) change after the first 15 min of tumbling and in different experimental conditions in fig. 2 experiments but they seem to evolve in fig. 3 experiments. Why is that? where pumices different? did the coloring affect their mechanical properties? in the lines 364-374 authors suggest that their textures could be different; if so, the trend should not be considered of general value.

Implications are: in a pdc deposit despite continuous ash production in the current lapilli morphology does not change/change from medial to distal locations. Please check for deposit data confirming this- what about the general shapes of clasts in real pdc/fall deposits?

-in the discussion, the authors compare results with granular flow- I see the effect of gas escape and ash elutriation as way less relevant than other concentrated (i.e. fluidised currents). Could they provide some example of observed deposits (for example, in recent eruptions of Lascar volcano).

-Fig. 4 and associated data are still part of the results obtained in the study and should be presented as such, not in the discussion chapter.

- While three orthogonal axes of the lapilli were measured for volume calculation, no 3d analysis was made for shape. This is a weakness of this study. I believe the authors have data to show shape variations on 3d Sneed and Folk classification plots. These plots should be added to the article.

Communications Earth & Environment is committed to improving transparency in authorship. As part of our efforts in this direction, we are now requesting that all authors identified as 'corresponding author' create and link their Open Researcher and Contributor Identifier (ORCID) with their account on the Manuscript Tracking System prior to acceptance. ORCID helps the scientific community achieve unambiguous attribution of all scholarly contributions. You can create and link your ORCID from the home page of the Manuscript Tracking System by clicking on 'Modify my Springer Nature account' and following the instructions in the link below. Please also inform all co-authors that they can add their ORCIDs to their accounts and that they must do so prior to acceptance.

If you experience problems in linking your ORCID, please contact the Platform Support Helpdesk.

Version 1:

Decision Letter:

Dear Ms Figueiredo,

Your manuscript titled "Shape evolution of pumice during granular flow" has now been seen by our reviewers, whose comments appear below. In light of their advice we are delighted to say that we are happy, in principle, to publish a suitably revised version in Communications Earth & Environment.

We therefore invite you to revise your paper one last time to address the remaining concerns of our reviewers. At the same time we ask that you edit your manuscript to comply with our format requirements and to maximise the accessibility and therefore the impact of your work.

EDITORIAL REQUESTS:

****Please take care to match our formatting and policy requirements. We will check revised manuscript and return manuscripts that do not comply. Such requests will lead to delays. ****

SUBMISSION INFORMATION:

OPEN ACCESS:

Communications Earth & Environment is a fully open access journal. Articles are made freely accessible on publication. For further information about article processing charges, open access funding, and advice and support from Nature Portfolio, please visit <https://www.nature.com/commsenv/open-access>

Link Redacted

Best regards,

Domenico Doronzo
Editorial Board Member
Communications Earth & Environment

Joe Aslin
Deputy Editor,
Communications Earth & Environment
Consulting Editor,
Communications Sustainability

<https://www.nature.com/commsenv/>
Twitter: @CommsEarth

REVIEWERS' COMMENTS:

Reviewer #1 (Remarks to the Author):

Dear editor,

in this manuscript, the authors address the shape and size evolution of pumice clasts during transport phases, using an approach based on tumbling experiments, evaluating a few morphometric parameters and the amount of ash generation. They compare a series of experimental settings to evaluate different conditioning factors of the studied problem independently, taking shape information in different time steps to analyze the timescale associated with this problem. The document is well-written and correctly organized, the presented figures are clear and support description of results and discussions, and conclusions are effectively based on the novel experimental results presented in the paper. I really appreciate the detailed responses to each of the reviewers' concerns, the modifications made to some figures, and the robust explanation of relevant limitations of this experimental approach. In particular, the comparison between transport conditions within the volcanic conduct and during PDC propagation significantly strengthens the manuscript, in my opinion, allowing the reader to identify better the natural scenario the authors are able to effectively reproduce and natural processes that are not included in the experimental setup.

Considering that all my previous concerns were successfully addressed, and despite I still have major doubts about the validity of comparing the effect of steel balls and lithic fragments, I support publication of this manuscript after minor editorial suggestions. In this sense, I am attaching a document with a series of small comments. Please not that my mother tongue is not English and some order mistakes were identified in Figures 2 and 5.

Alvaro Aravena

Reviewer #2 (Remarks to the Author):

Dear Authors

Thank you for robustly addressing all concerns discussed. The responses and adjustments made to the manuscript appear adequate for it to progress. The manuscript presents data and results that may be of interest to researchers studying experimental flow processes. While 2D measures of clasts seem appropriate, it is my opinion that these measures are dated (and difficult to relate) and can easily and more efficiently be replaced by 3D techniques but you have provided some justification on your choice of methods. The breadth of factors that influence natural flows makes it hard to scale this type of test but the manuscript may be of interest to the broader community.

I look forward to seeing the manuscript published.

Thank you

** Visit Nature Portfolio's author and referees' website at www.nature.com/authors for information about policies, services and author benefits**

Shape evolution of pumice during granular flow

Carolina Figueiredo^{1*}, Ulrich Kueppers¹, Lisa Depauli¹, Sarp Esenyel¹, Luiz
Pereira¹, Donald B. Dingwell.¹

¹ Department of Earth and Environmental Sciences, Ludwig-Maximilians-
Universität München, 80333 Munich, Germany.

ORCID (CF): 0000-0003-1162-0240

ORCID (UK): 0000-0003-2815-1444

ORCID (LP): 0000-0001-9555-0352

ORCID (DBD): 0000-0002-3332-789X

*Corresponding authors: figueiredo.ca@lmu.de

Keywords: pyroclasts, pyroclastic density current, abrasion, explosive eruption,
relaxation theory.

Abstract

Explosive volcanic eruptions are a major geo-hazard whose lethality grows daily with
human population. Given the energetic nature of eruptive processes, direct
observation of key phenomena is limited or impossible. Accordingly, the study of
volcanic deposits and pyroclast textures are essential for understanding eruption and
transport processes. In parallel, experimental constraints increasingly provide a vital
contribution to improving hazard assessment.

We performed three types of tumbling experiments using pumice lapilli from the 13ka
Laacher See eruption (Eifel, Germany) to investigate volcanic ash generation and
pyroclast shape evolution. To this end, samples were sieved and two petrophysical
(volume, porosity) as well as four morphological parameters (Axial Ratio, Convexity,
Form Factor, Solidity) of 100-clasts were measured before start and after five
experimental steps. We observed the strongest shape change in the first 15 minutes
and up to 50wt.% of ash generation.

We frame our analysis in terms of effective relaxation timescales, whereby pyroclasts
display a decelerating rate of change towards a time-invariant shape. This
quantification of the susceptibility of porous pyroclasts to shape and size changes
enhances our understanding of transport processes. The related empirical law will
allow volcanologists to assess the shape- and time-dependent response to mechanical
exposure during transport.

1. Introduction

During the explosive eruption of felsic magma, fragmentation is dominated by brittle
failure^{1,2}. The efficiency of this process depends on textural characteristics (e.g.,
porosity, crystallinity, permeability) and the transient gas overpressure in pore space,
resulting from the competition between degassing and outgassing. Upon
fragmentation, angular porous pyroclasts of variable grain size are accelerated and
eventually ejected into the atmosphere^{3,4}. The eruptive and transport dynamics control

the final distribution of the pyroclastic deposits. Fall (gravitational settling dominated)
and flow (sub-horizontal transport) ~~deposits~~ are the two end member deposit types of
such explosive events. The meaning of the grain size distribution as revealed from
volcanic deposits is a matter of discussion as it may depend – at variable degrees - on
primary (magma fragmentation) and post-fragmentation transport processes.

Fall deposits tend to ~~be characterised by~~ the effects of transport-related sorting due to
atmospheric conditions (e.g., wind field, humidity) as well as particle density and
aerodynamic properties⁵⁻⁸. In medial to distal regions, such deposits commonly contain
predominantly angular lapilli-sized clasts and exhibit evidence of density- and/or grain
size-sorting. In contrast, pyroclastic density currents (PDCs) advance rapidly down
volcanic flanks, experiencing both density-stratification and turbulence. Particle
concentration and flow turbulence define a wide spectrum of these currents, ranging
from particle-poor (pyroclastic surges) to granular flows with high granular
temperature. ~~In the latter,~~ particle-particle and particle-substrate interactions are
anticipated to be ubiquitous⁹⁻¹². Porous pyroclasts accordingly undergo evolution in
size and shape¹³⁻¹⁵, associated with the generation of fine particles, including those
<2mm, termed volcanic ash. The ash content, either part of a flow from the beginning
or formed in-situ by collisional/frictional processes, affects the flow's mobility, enlarging
the areas at threat.

The efficiency and impact of the experimental conditions on such size and shape
evolution of pyroclasts ~~has~~ been investigated experimentally via (1) tumbling¹⁶⁻¹⁹ and
(2) drop^{20,21} experiments as well as with the aid of numerical models¹⁴. Clast abrasion
and morphology evolution have also been evaluated for gravel particles in dry and wet
settings, relevant to ~~debris flows~~^{22,23} and bed load transport in rivers²⁴. Those studies
indicate a strong influence of experimental conditions on clast shape evolution and the
resulting dynamic coupling to the surrounding medium²⁵. ~~Thus, in principle,~~
~~experiments are capable of accessing the variation of morphological parameters.~~
Independent of the experimental conditions and material properties, a widely
recognised principle of particle abrasion remains which states that if a particle does
not fracture anew during the process, then its tendency to change shape decreases
over time eventually reaching a time-invariant 'equilibrium state' in terms of shape and
roughness²⁶.

The physical concept of relaxation was initially proposed to describe the dynamics of
gases and applied to dielectric materials. Nowadays, it has been used across various
systems, including glass science, superconductors, jamming in granular materials as
well as biological and magnetic systems^{1,27-30}. It is vastly used to study structural
relaxation of glasses, whereby a ~~structure~~ and consequently the material properties
evolve to equilibrium at a rate dictated by a characteristic relaxation time^{31,32}. During
relaxation, the rate of change of a property commonly decelerates as this property
approaches the equilibrium or optimal state³¹. Here we investigate the shape relaxation
of abrading lapilli using laboratory experiments.

2. Results

We have conducted tumbling experiments under **controlled conditions** and quantified
the related abrasion process in terms of the ash fraction and a series of morphological
parameters of particle shape. We have analysed the observed fine particle generation
(section 2.1) and clast shape change (section 2.2) in terms of relaxation times for
shape and roughness evolution.

2.1 Ash generation

We present the results of dry tumbling experiments conducted at room temperature
and ambient pressure, expanding an empirical correlation presented by Hornby et al.¹⁷.
We used pumice lapilli from Laacher See (East Eifel, Germany; sourced from fall
deposits of the 13 ka phonolithic Plinian Laacher See eruption near Nickenich,
Germany). Stratigraphically, they belong to the ~~units~~ Middle (MLST) and Upper (ULST)
Laacher See Tephra and contain 10-15 vol.% (dense rock equivalent) **phenocrysts**³³.

A strong non-linear relationship between tumbling duration (displayed as tumbling
distance – calculated using drum diameter and experiment duration¹⁷) and ash
generated was observed for three experimental conditions of the so-called **Set A (T1,
T2, T3B) and one for Set B (T1) experiments**. The ash produced was quantified as the
weight fraction of experimentally generated volcanic ash (based on the starting weight)
and is displayed cumulatively in Fig. 1.

Ash generation was highest at the beginning of the experiments (within the first 1000
meters of tumbling) and decreased non-linearly with increasing ~~further~~ experimental
duration. This trend can be visualised as a “relaxation” of ash generation efficiency,
dependent on the bulk contact energy per experiment type (see Methods). A sense of
the magnitude of these changes in our experiments is provided by the observation that
amongst all experiments, T3B (i.e., ash removed at each time step and presence of
steel balls) produced the largest amount of ash (up to 48 wt.%), followed by T2 (i.e.,
ash removed at each time step, ca. 36 wt.%), and T1 (i.e., ash returned to the drum
after each time step, ca. 25 wt.%). These results document the prime influence of bulk
contact energy on ash generation. The presence of inter-clast ash led to reduced ash
generation (T1 compared to T2) while the steel balls locally increased point loads that
lead to abrasion (dominant) or break-up (T3 compared to T2).

Figure 1: Ash production during tumbling experiments, plotted as cumulative ash fraction (weight fraction of the starting mass) against the rotational distance for experiments T1, T2 and T3B (*Set A*) and T1 (*Set B*). Result reproducibility is being shown by 2 data sets each (this study) and in accordance with earlier studies¹⁷. *Set B* experiment T1 with coloured clasts reproduces the ash generation trend, proofing the subordinate impact of the food colouring on the mechanical properties of the 100 coloured clasts.

2.2 Shape and petrophysical analysis

We have extended the analysis and have evaluated 2 petrophysical (volume and
 porosity) and 4 morphological (*Axial Ratio* [particle elongation], *Convexity* [fine-scale
 textural roughness], *Form Factor* [overall particle irregularity] and *Solidity*
 [morphological roughness]) parameters that describe the shape evolution during our
 experiments. We have cast the analysis of these experimental results in terms of
 relaxation theory, yielding a novel insight into particle abrasion. Key parameters that
 govern the evolution of volcanic pyroclastic particle shapes are generated, providing a
 useful way for volcanologists to understand transport and sedimentation processes
 (and thereby eruption dynamics) based on thorough analysis of deposits.

[revised manuscript text omitted]

The experience gained from the experiments above led us to conclude that a higher
temporal resolution was required to effectively parameterise shape evolution. This was
addressed by designing *Set B* experiments. Here, 100 clasts, dyed with food colour
could be sampled repeatedly after each experimental increment to capture detailed
trends in volume reduction, visualised as changes of *Convexity*, *Form Factor*, *Axial*
*Ratio* and *Solidity* (Fig. 3). The refined temporal trends confirm the strongly nonlinear
~~correlation~~ and indicate that measurable shape changes ensue very rapidly during
transport (Fig. 4b). The observation that the number of coloured starting clasts (n=100)
increased to 106 (three lapilli broke apart) and 120 (ten lapilli broke apart) clasts after
20 and 60 minutes, respectively, underscores the presence but not the dominance of
clast breakage as source of ash particles during tumbling. The breakage of these clasts
was not significant enough to cause discontinuities or abrupt changes in the trends
(Fig. 3), indicating that particle abrasion is the dominant mechanism driving shape
evolution during the tumbling experiments.

Granular materials exhibit variability in both their morphological properties, such as
 grain shapes, size distributions, and surface characteristics; and their microstructural
 arrangements, including the organisation of grains and their contact points³⁷. In our
 tumbling experiments, we do not address the microstructural arrangements due to the
 complexity and variability of how grains organise and interact at a small scale.
 Interactions between particles of dry, rough, and granular materials have been
 experimentally defined as being mainly due to friction and collision³⁸, and
 fragmentation and abrasion are the main processes that govern disruption of the
 frictional materials³⁹. In collisional interactions with a significant overlap of clast cross-
 sectional area (>50%), particle break-up is promoted, whereas in the tangential case

Figure 4: Axial ratio vs. Form Factor plots for Set A experiments (A) and Set B experiments (B). The dotted black line indicates the limit between SM and tumbled clasts. The Kernell density functions to the right and the top of both plots distinctly allow for tracing the temporal shape evolution of the Set B clasts. Shadowgraphs of the specific clasts (B) show the decrease of surface roughness comparing starting material and tumbled clasts.

with overlap (<10%), impact energy is predominantly translated into rotational movement, yielding abrasion. Based on the here observed propensity of porous pyroclasts to shape change, we conclude from the angular lapilli clasts observed in fall deposits of explosive eruptions indicate that collisional (promoting disruption) and/or tangential (promoting abrasion) interactions with other pyroclasts or the conduit walls are of subordinate importance there.

Our experimentally derived data clearly demonstrate the high potential for porous pyroclasts to quickly (within minutes) change their shape. During most explosive volcanic eruptions, porous magma (per definition containing a melt fraction) is fragmented in a brittle way, generating angular fragments of variable size. Depending on clast temperature and melt composition, little or no (substantial) shape alteration driven by surface tension occurs after such

fragmentation event. Fall deposits of such explosive eruptions are commonly
dominated by angular clasts. The laboratory experiments presented here are less
energetic than those extant during volcanic currents (grain flows, PDCs), yet particle
shapes have adjusted very rapidly. During magmatic fragmentation, magmatic volatiles
at overpressure force the rising and deforming magma (viscous behaviour) into (local)
brittle response to the acting stress. The liberated gas volume will expand according
to the highest pressure differential (in most cases towards Earth surface) and
aerodynamic drag will accelerate the surrounding pyroclasts. Laboratory experiments
mimicking explosive eruptions have shown that (1) particle exit velocity strongly decays
with time⁴⁰ (this does not refer to particle deceleration) and (2) particles of any grain
size travel at comparable velocities (up to 250 m/s with velocity differences rarely
exceeding few m/s;³ and unpublished data) when leaving the experimental vent. We
assume a similar dilatational flow field inside volcanic conduits from above the
fragmentation level all the way to the surface. If this holds true, pyroclast-pyroclast
collisions are unlikely or of low impact energy.

The effect of achieving an equilibrium of the smoothness of a surface is documented
by the established plateaus of *Convexity*, *Form Factor*, and *Solidity* (Fig. 2 and 3),
regardless of the experiment type. This phenomenon has been termed “equilibrium
roughness” in the field of tribology²⁶. It depicts how surfaces adapt during initial contact
under sliding conditions to reach a state of minimum wear and friction.

Abrasion efficiency is interpreted as the rate of roughness change, and (neglecting
minor contributions from further fragmentation) the sample can be treated as an
approximation of a fluid continuum with roughness as an internal variable directly
affected by viscosity, which can be defined as a measure of a fluid’s resistance to
flow^{39,41-46}. Turbulent fluctuations create an energy cascade process, where the stress
power from the mean flow dissipates across characteristic length scales of the flow.
Literature contains several examples of complex equations dependent of several
material properties used to model granular materials³⁷. However, there is a lack of
information on how the shape of these materials changes over time and of the
implications of this in the flow dynamics. In this work we introduce a simple equation
with three fitting (t, β, p_∞) parameters to explain the shape evolution of frictional
granular materials.

We propose that the equilibrium roughness phenomenon can be described by a
relaxation law^{29,30}. Perhaps the best studied example of the application of relaxation
theory in volcanology is its application to the liquid-to-glass transition^{1,2}. It has also
been applied to the topic of viscous shape relaxation of vesicles and its impact on the
suspension rheology of vesicular magmas⁴⁷⁻⁴⁹.

In the present context, as considering the flowing granular material as a fluid
continuum, we can treat the shape parameters - *Convexity*, *Form Factor*, and *Solidity*
- in terms of relaxation times within the flow. These parameters account for large-scale
and small-scale roughness and form, respectively, whereby each exhibit, in principle,
a specific relaxation time.

When a material undergoes a certain change, any feature of physical property at a
given time $p(t)$ evolves with aging time from its initial value p_0 to its final and value p_∞ ,
such that a normalised relaxation function $\phi(t)$ is the following:

$$329 \quad \phi(t) = \frac{p(t) - p_\infty}{p_0 - p_\infty}. \quad [\text{eq. 1}]$$

We employ the Kohlrausch-Williams-Watts (KWW) equation (eq. 2) that describes the
temporal function decay of a given property p ^{29,30}:

$$332 \quad \phi(t) = \exp \left[- \left(\frac{t}{\tau_k} \right)^\beta \right], \quad [\text{eq. 2}]$$

where τ_k is the characteristic relaxation time, and β describes the breadth of the
relaxation time distribution, serving as the non-exponentiality parameter. In case β is
unity, it corresponds to a simple case of exponential decay, representing a single
relaxation time, i.e., the process happens all over the material at the same rate. For 0
$< \beta < 1$, it indicates a stretched exponential decay, meaning a distribution of relaxation
338 times, where β close to unity represents a narrow distribution, while β close to zero
represents a broad range of relaxation times across the material. A combination of
equations eq.1 and eq.2 gives rise to:

$$341 \quad p(t) = p_\infty + (p_0 - p_\infty) \exp \left[- \left(\frac{t}{\tau_k} \right)^\beta \right] \quad [\text{eq. 3}]$$

We used the mean values of *Convexity*, *Form Factor*, and *Solidity* for each time
increment (0, 5, 10, 15, 20, and 60 minutes) of experiment T1 (*Set B*) as the studied
property p (Fig. 5). Time increment 0 refers to the starting material (SM). The relaxation
timescale for *Convexity* is 5.1 minutes (Fig. 5A), *Form Factor* 8.7 minutes (Fig. 5B) and
*Solidity* 12.5 minutes (Fig. 5C), respectively, implying that finer-scale adjustments
occur on shorter timescales than changes to the overall shape and structural
roughness of the particles.

By analysing eq. 3, τ_k refers to the time required for particles relaxation 63.2 % of its
path to equilibrium if $\beta = 1$. However, for this studied case, β is not unity (see Fig. 5).
Therefore, for our studied case instead of 63.2 % we obtained 42 %, 50 % and 43 %
of its path to equilibrium for *Convexity*, *Form Factor*, and *Solidity*, respectively. As
mentioned, β reflects the distribution of relaxation times, which in glass science is
linked to the varying structures on atomic to nanoscale levels⁵⁰⁻⁵², therefore these
different regions undergo relaxation following different rates. By analogy with the
current system, we suggest that the β values (cf. Fig. 5) illustrate how different particles
undergo shape relaxation and the close it is to unity means that particles change shape
in the same rate (narrow distribution of shape relaxation time), while for β going
towards zero, means a broad distribution of shape relaxation times. The analysis of
this parameter can be useful tool to evaluate how heterogeneous our set of particles
is. Interestingly, comparing the experiments with and without particle tracking (using
food colouring) yielded different results, characterising our system with β values not

Figure 5: Temporal evolution of particle shape parameters during tumbling experiment type T1. The plots depict the variation of (A) Solidity, S , (B) Convexity, C , and (C) Form Factor, FF , over time t [min]. The solid black lines represent the best fit to the experimental data (red dots) using the relaxation model described by equation [1], and the initial (S_0) and final values are provided within each plot.

403

near unity, as different particles change shape in different rates (cf. Fig. 2 and Fig. 3). These variations might be related to different textures of the starting materials, such as crystal and bubble content and shape.

The parameterisation of the evolution of pyroclast morphology presented here enhances our physical understanding of transport of pyroclastic materials, in detail the efficiency of particle-particle interaction and related ash generation. We believe this to be of great significance as collisional processes take place in a wide variety of volcanic settings. Pyroclasts formed by brittle fragmentation and preserved as angular fragments in fall deposits serve as proxy for natural processes and serve as reference point against which pyroclasts collected from PDC deposits may be compared. We suggest the use of relaxation theory to describe the shape evolution of particle in terms of *Convexity, Form Factor and Solidity*. Thorough quantitative analysis of clasts from primary volcanic deposits directly in the field will avoid any post-sampling size and shape alteration. We propose that this will allow for upscaling the empirical laws from laboratory experiments and contribute to enhanced understanding of transportation processes, in particular those affecting the mobility of density

currents. Eventually, this will serve to educate scientists advising the population on and
 around volcanoes and ultimately reduce the lethality of such terrible events.

**4. Methods**

Rotary tumbling experiments that mimic particle interactions during PDCs or granular
 flows have been used to study the abrasion of pumice lapilli. The particle interactions,
 albeit likely of lesser intensity than possible in natural transport, are caused by forced
 tumbling of the experimental charge. Previous studies¹⁷⁻¹⁹ have empirically constrained
 the relationship between tumbling conditions and ash generation. To increase our
 mechanistic understanding of pyroclast size and shape evolution during PDC
 transport, we conducted further tumbling experiments as defined below. We extended
 the analysis of the experimental cargo and have evaluated two petrophysical (volume
 and porosity) and four morphological (*Axial Ratio* [particle elongation], *Convexity* [fine-
 scale textural roughness], *Form Factor* [overall particle irregularity] and *Solidity*
 [morphological roughness]) parameters that describe the shape evolution during our
 experiments.

4.1 Tumbling experiments and ash generation

The tumbling experiments were carried out at ambient conditions in an industrial
 cement mixer (see Hornby et al.¹⁷ for detailed description). The starting material (SM)
 was sourced from fall deposits of the 13 ka phonolithic Plinian Laacher See eruption
 near Nickenich, Germany. Each experiment started with 2 kg of pumice lapilli
 (n~3.000) with a porosity range of 45-93% -mean porosity 80%- and particle size of 2-

Set	Experiment type	Description	Duration	Target clasts
A	T1	Ash and lapilli returned to drum in each step for further tumbling	15', 30', 45', 60', 120' 	random (n=100)
	T2	Only lapilli returned to drum, ash stored separately in each step		
	T3B	T2 + 2 steel balls (220g each)		
B	T1	Ash and lapilli returned to drum in each step for further tumbling	5', 10', 15', 20', 60'	coloured (n=100)

Table 1: Description of the experimental conditions of two sets and three types of tumbling experiments. Collisional energy was varied by removing/keeping ash from/in the drum after each increment or adding steel balls. Clasts have been investigated after five time-steps (representing theoretical tumbling distance). In *Set A*, random clasts were selected for analysis, in *Set B*, the same batch of 100 clasts was investigated after each increment.

5 cm. In *Set A*, three types of experiments were performed to map out abrasion
 efficiency as a function of collisional energy. In T1 experiments, lapilli and ash
 generated were put back into the drum after each time increment. In T2, only the
 remaining lapilli fraction was put back into the drum, the ash was sampled and stored.
 In T3B, we added two steel balls to mimic the frequent presence of higher-density (than
 pumice) clasts during natural transport. Only the lapilli fraction was put back into the
 drum after each time increment. Ash generation was quantified by dry sieving at 2 mm
 after 15, 30, 45, 60 and 120 minutes, respectively (Table 1).

For T1, we performed a second set of experiments (*Set B*), in which 100 lapilli from
 within the 2 kg starting material had been coloured with food colour prior to tumbling.

This treatment did not affect the mechanical response as density and weight remained
 effectively unchanged. Ash generation was quantified by dry sieving at 2 mm after 5,
 10, 15, 20 and 60 minutes, respectively.

4.2 Petrophysical properties and shape analysis

In addition to obtaining the weight fraction of ash generated per experimental
 increment, we also quantified clast volume (Fig. 6a), weight and morphology (Fig. 6b).
 Clast volumes were calculated using the manual method⁵³, measuring the length of
 three orthogonal axes with a calliper and calculating the volume assuming a three-axial
 ellipsoid geometry. High-precision weight analysis (10^{-4} g) allowed for calculating the
 porosity of each clast. For shape analysis, we used the projected area of the clasts
 from digital photographs, processed the images on Photoshop (clast contours
 delimitation and binarization) and calculated shape parameters on *ImageJ*. We
 increased the resolution compared to earlier studies¹⁷ by analysing macro images
 taken of each clast individually (and not in sets of $n=100$). We selected *Axial Ratio*,
 *Convexity*, *Form Factor* and *Solidity*⁵⁴ and used a adapted shape macro⁵⁵. In *Set A*,
 100 randomly selected clasts were taken from the lapilli fraction for analysis and
 afterwards put back to the lapilli cargo in the drum for the subsequent tumbling. In *Set B*,
 clasts coloured with food colour could be picked after each experimental increment
 for analysis and afterwards put back to the lapilli cargo in the drum for the subsequent
 tumbling.

Figure 6: Workflow to constrain the pumice clasts petrophysical properties, volume (a) and shape (b). a) Calliper measurement of three orthogonal axes (A, B, and C) and volume calculation. B) Illustration of the parametrization measurements (*Axial Ratio*, *Convexity*, *Form Factor* and *Solidity*) on a binary image of pumice clast.

Acknowledgements

CF and UK acknowledge support by grant KU2689/7-1 from the Deutsche
 Forschungsgemeinschaft. LP is grateful for the support of the Alexander von Humboldt
 Foundation. DBD, LP, & CF acknowledge the support of ERC 2018 ADV Grant 834225
 (EAVESDROP) to DBD.

References

- Dingwell, D. B. & Webb, S. L. Structural relaxation in silicate melts and non-Newtonian
 melt rheology in geologic processes. *Physics and Chemistry of Minerals* **16** (1989).
 <https://doi.org/10.1007/bf00197020>

Dingwell, D. B. Volcanic Dilemma--Flow or Blow? *Science* **273**, 1054-1055 (1996).
<https://doi.org/10.1126/science.273.5278.1054>

Kueppers, U., Perugini, D. & Dingwell, D. B. "Explosive energy" during volcanic
eruptions from fractal analysis of pyroclasts. *Earth and Planetary Science Letters* **248**,
800-807 (2006). <https://doi.org/10.1016/j.epsl.2006.06.033>

Spieler, O. *et al.* The fragmentation threshold of pyroclastic rocks. *Earth and Planetary*
*Science Letters* **226**, 139-148 (2004). <https://doi.org/10.1016/j.epsl.2004.07.016>

Carey, S. N. & Sigurdsson, H. Influence of particle aggregation on deposition of distal
tephra from the M_{Ay} 18, 1980, eruption of Mount St. Helens volcano. *Journal of*
*Geophysical Research: Solid Earth* **87**, 7061-7072 (1982).
<https://doi.org/10.1029/JB087iB08p07061>

6 Pyle, D. M. The thickness, volume and grainsize of tephra fall deposits. *Bulletin of*
*Volcanology* **51**, 1-15 (1989). <https://doi.org/10.1007/bf01086757>

7 Sparks, R. S. J., Bursik, M. I., Ablay, G. J., Thomas, R. M. E. & Carey, S. N. Sedimentation
of tephra by volcanic plumes. Part 2: controls on thickness and grain-size variations of
tephra fall deposits. *Bulletin of Volcanology* **54**, 685-695 (1992).
<https://doi.org/10.1007/bf00430779>

Wilson, L. & Huang, T. C. The influence of shape on the atmospheric settling velocity
of volcanic ash particles. *Earth and Planetary Science Letters* **44**, 311-324 (1979).
[https://doi.org/10.1016/0012-821x\(79\)90179-1](https://doi.org/10.1016/0012-821x(79)90179-1)

Dufek, J. & Bergantz, G. W. Suspended load and bed-load transport of particle-laden
gravity currents: the role of particle–bed interaction. *Theoretical and Computational*
*Fluid Dynamics* **21**, 119-145 (2007). <https://doi.org/10.1007/s00162-007-0041-6>

Jones, T. J., Shetty, A., Chalk, C., Dufek, J. & Gonnermann, H. M. Identifying rheological
regimes within pyroclastic density currents. *Nature Communications* **15**, 4401 (2024).
<https://doi.org/10.1038/s41467-024-48612-7>

Taddeucci, J. & Palladino, D. Particle size-density relationships in pyroclastic deposits:
inferences for emplacement processes. *Bulletin of Volcanology* **64**, 273-284 (2002).
<https://doi.org/10.1007/s00445-002-0205-6>

Valentine, G. A. Stratified flow in pyroclastic surges. *Bulletin of Volcanology* **49**, 616-
630 (1987). <https://doi.org/10.1007/bf01079967>

13 Buckland, H. M., Eychenne, J., Rust, A. C. & Cashman, K. V. Relating the physical
properties of volcanic rocks to the characteristics of ash generated by experimental
abrasion. *Journal of Volcanology and Geothermal Research* **349**, 335-350 (2018).
<https://doi.org/10.1016/j.jvolgeores.2017.11.017>

14 Dufek, J. & Manga, M. In situ production of ash in pyroclastic flows. *Journal of*
*Geophysical Research: Solid Earth* **113** (2008). <https://doi.org/10.1029/2007jb005555>

15 Walker, G. P. L. Generation and dispersal of fine ash and dust by volcanic eruptions.
*Journal of Volcanology and Geothermal Research* **11**, 81-92 (1981).
[https://doi.org/10.1016/0377-0273\(81\)90077-9](https://doi.org/10.1016/0377-0273(81)90077-9)

16 Cagnoli, B. & Manga, M. Granular mass flows and Coulomb's friction in shear cell
experiments: Implications for geophysical flows. *Journal of Geophysical Research:*
*Earth Surface* **109** (2004). <https://doi.org/10.1029/2004jf000177>

17 Hornby, A., Kueppers, U., Maurer, B., Poetsch, C. & Dingwell, D. Experimental
constraints on volcanic ash generation and clast morphometrics in pyroclastic density
currents and granular flows. *Volcanica* **3**, 263-283 (2020).
<https://doi.org/10.30909/vol.03.02.263283>

- 18 Kueppers, U., Putz, C., Spieler, O. & Dingwell, D. B. Abrasion in pyroclastic density
currents: Insights from tumbling experiments. *Physics and Chemistry of the Earth, Parts*
*A/B/C* **45-46**, 33-39 (2012). <https://doi.org/10.1016/j.pce.2011.09.002>
- 19 Manga, M., Patel, A. & Dufek, J. Rounding of pumice clasts during transport: field
measurements and laboratory studies. *Bulletin of Volcanology* **73**, 321-333 (2011).
<https://doi.org/10.1007/s00445-010-0411-6>
- 20 Mueller, S. B., Lane, S. J. & Kueppers, U. Lab-scale ash production by abrasion and
collision experiments of porous volcanic samples. *Journal of Volcanology and*
*Geothermal Research* **302**, 163-172 (2015).
<https://doi.org/10.1016/j.jvolgeores.2015.07.013>
- 21 Schwarzkopf, L. M., Spieler, O., Scheu, B. & Dingwell, D. B. Fall-experiments on Merapi
basaltic andesite and constraints on the generation of pyroclastic surges. *eEarth* **2**, 1-
5 (2007). <https://doi.org/10.5194/ee-2-1-2007>
- Caballero, L., Sarocchi, D., Borselli, L. & Cárdenas, A. I. Particle interaction inside debris
flows: Evidence through experimental data and quantitative clast shape analysis.
*Journal of Volcanology and Geothermal Research* **231-232**, 12-23 (2012).
<https://doi.org/10.1016/j.jvolgeores.2012.04.007>
- Yao, T., Yang, H., Lourenço, S. D. N., Baudet, B. A. & Kwok, F. C. Y. Multi-scale particle
morphology evolution in rotating drum tests: Role of particle shape and pore fluid.
*Engineering Geology* **303** (2022). <https://doi.org/10.1016/j.enggeo.2022.106669>
- Lewin, J. & Brewer, P. A. Laboratory simulation of clast abrasion. *Earth Surface*
*Processes and Landforms* **27**, 145-164 (2002). <https://doi.org/10.1002/esp.306>
- Deal, E. *et al.* Grain shape effects in bed load sediment transport. *Nature* **613**, 298-302
(2023). <https://doi.org/10.1038/s41586-022-05564-6>
- Kragelsky, I. V. (eds I.V. Kragelsky, M.N. Dobyichin, & V.S. Kombalov) 297–316
(Pergamon, 1982).
- Campbell, I. A. & Giovannella, C. Relaxation in Complex Systems and Related Topics.
*Springer* (1990). <https://doi.org/10.1007/978-1-4899-2136-9>
- Jaeger, H. M. Celebrating Soft Matter's 10th Anniversary: Toward jamming by design.
*Soft Matter* **11**, 12-27 (2015). <https://doi.org/10.1039/c4sm01923g>
- Kohlrausch, R. Theorie des elektrischen Rückstandes in der Leidener Flasche. *Annalen*
*der Physik* **167**, 179-214 (1854). <https://doi.org/10.1002/andp.18541670203>
- Williams, G. & Watts, D. C. Non-symmetrical dielectric relaxation behaviour arising
from a simple empirical decay function. *Transactions of the Faraday Society* **66** (1970).
<https://doi.org/10.1039/tf9706600080>
- Lancelotti, R. F., Zanotto, E. D. & Sen, S. Kinetics of physical aging of a silicate glass
following temperature up- and down-jumps. *The Journal of Chemical Physics* **160**
(2024). <https://doi.org/10.1063/5.0185538>
- Lancelotti, R. F., Pereira, L., Hess, K.-U., Dingwell, D. B. & Zanotto, E. D. Flash-DSC
provides valuable insights into glass relaxation and crystallization. *Journal of Non-*
*Crystalline Solids* **646** (2024). <https://doi.org/10.1016/j.jnoncrysol.2024.123242>
- Schmincke, H.-U., Park, C. & Harms, E. Evolution and environmental impacts of the
eruption of Laacher See Volcano (Germany) 12,900 a BP. *Quaternary International* **61**,
61-72 (1999). [https://doi.org/10.1016/s1040-6182\(99\)00017-8](https://doi.org/10.1016/s1040-6182(99)00017-8)
- Cleary, P. W. DEM simulation of industrial particle flows: case studies of dragline
excavators, mixing in tumblers and centrifugal mills. *Powder Technology* **109**, 83-104
(2000). [https://doi.org/10.1016/s0032-5910\(99\)00229-6](https://doi.org/10.1016/s0032-5910(99)00229-6)

- Freundt, A. & Schmincke, H. U. Abrasion in pyroclastic flows. *Geologische Rundschau*
**81**, 383-389 (1992). <https://doi.org/10.1007/bf01828605>
- Breard, E. C. P. *et al.* The fragmentation-induced fluidisation of pyroclastic density
currents. *Nature Communications* **14** (2023). [https://doi.org/10.1038/s41467-023-](https://doi.org/10.1038/s41467-023-37867-1)
[37867-1](https://doi.org/10.1038/s41467-023-37867-1)
- Radjai, F., Roux, J.-N. & Daouadji, A. Modeling Granular Materials: Century-Long
Research across Scales. *Journal of Engineering Mechanics* **143** (2017).
[https://doi.org/10.1061/\(asce\)em.1943-7889.0001196](https://doi.org/10.1061/(asce)em.1943-7889.0001196)
- Yu, F. *et al.* Particle breakage of sand subjected to friction and collision in drum tests.
*Journal of Rock Mechanics and Geotechnical Engineering* **13**, 390-400 (2021).
<https://doi.org/10.1016/j.jrmge.2020.08.004>
- Kirchner, N. P. Thermodynamically consistent modelling of abrasive granular materials.
I Non-equilibrium theory. *Proceedings of the Royal Society of London. Series A:*
*Mathematical, Physical and Engineering Sciences* **458**, 2153-2176 (2002).
<https://doi.org/10.1098/rspa.2002.0963>
- Cigala, V. *et al.* The dynamics of volcanic jets: Temporal evolution of particles exit
velocity from shock-tube experiments. *Journal of Geophysical Research: Solid Earth*
**122**, 6031-6045 (2017). <https://doi.org/10.1002/2017jb014149>
- Daniel, R. C., Poloski, A. P. & Eduardo Sáez, A. A continuum constitutive model for
cohesionless granular flows. *Chemical Engineering Science* **62**, 1343-1350 (2007).
<https://doi.org/10.1016/j.ces.2006.11.035>
- Fang, C., Wang, Y. & Hutter, K. A thermo-mechanical continuum theory with internal
length for cohesionless granular materials. *Continuum Mechanics and*
*Thermodynamics* **17**, 545-576 (2006). <https://doi.org/10.1007/s00161-006-0007-8>
- Fang, C., Wang, Y. & Hutter, K. Shearing flows of a dry granular material—hypoplastic
constitutive theory and numerical simulations. *International Journal for Numerical and*
*Analytical Methods in Geomechanics* **30**, 1409-1437 (2006).
<https://doi.org/10.1002/nag.525>
- Fang, C. & Wu, W. On the weak turbulent motions of an isothermal dry granular dense
flow with incompressible grains: part I. Equilibrium turbulent closure models. *Acta*
*Geotechnica* **9**, 725-737 (2014). <https://doi.org/10.1007/s11440-014-0313-4>
- Göncü, F. & Luding, S. Effect of particle friction and polydispersity on the macroscopic
stress-strain relations of granular materials. *Acta Geotechnica* **8**, 629-643 (2013).
<https://doi.org/10.1007/s11440-013-0258-z>
- Volfson, D., Tsimring, L. S. & Aranson, I. S. Partially fluidized shear granular flows:
continuum theory and molecular dynamics simulations. *Phys Rev E Stat Nonlin Soft*
*Matter Phys* **68**, 021301 (2003). <https://doi.org/10.1103/PhysRevE.68.021301>
- Bagdassarov, N. S. & Dingwell, D. B. A rheological investigation of vesicular rhyolite.
*Journal of Volcanology and Geothermal Research* **50**, 307-322 (1992).
[https://doi.org/10.1016/0377-0273\(92\)90099-y](https://doi.org/10.1016/0377-0273(92)90099-y)
- Bagdassarov, N. S. & Dingwell, D. B. Frequency dependent rheology of vesicular
rhyolite. *Journal of Geophysical Research: Solid Earth* **98**, 6477-6487 (1993).
<https://doi.org/https://doi.org/10.1029/92JB02690>
- Bagdassarov, N. S. & Dingwell, D. B. Deformation of foamed rhyolites under internal
and external stresses: an experimental investigation. *Bulletin of Volcanology* **55**, 147-
154 (1993). <https://doi.org/10.1007/bf00301512>

- Cormier, L., Galois, L., Lelong, G. & Calas, G. From nanoscale heterogeneities to
nanolites: cation clustering in glasses. *Comptes Rendus. Physique* **24**, 199-214 (2023).
<https://doi.org/10.5802/crphys.150>
- Greaves, G. N. EXAFS and the structure of glass. *Journal of Non-Crystalline Solids* **71**,
203-217 (1985). [https://doi.org/10.1016/0022-3093\(85\)90289-3](https://doi.org/10.1016/0022-3093(85)90289-3)
- Le Losq, C. *et al.* Percolation channels: a universal idea to describe the atomic structure
and dynamics of glasses and melts. *Scientific Reports* **7** (2017).
<https://doi.org/10.1038/s41598-017-16741-3>
- Pisello, A. *et al.* The porosity of felsic pyroclasts: laboratory validation of field-based
approaches. *Bulletin of Volcanology* **85**, 69 (2023). [https://doi.org/10.1007/s00445-](https://doi.org/10.1007/s00445-023-01679-4)
[023-01679-4](https://doi.org/10.1007/s00445-023-01679-4)
- Liu, E. J., Cashman, K. V. & Rust, A. C. Optimising shape analysis to quantify volcanic
ash morphology. *GeoResJ* **8**, 14-30 (2015). <https://doi.org/10.1016/j.grj.2015.09.001>
- Figueiredo, C. A., Bongiolo, E. M., Jutzeler, M., da Fonseca Martins Gomes, O. &
Neumann, R. Alkaline pyroclast morphology informs on fragmentation mechanisms,
Trindade Island, Brazil. *Journal of Volcanology and Geothermal Research* **428** (2022).
<https://doi.org/10.1016/j.jvolgeores.2022.107575>

**Author contributions.** CF and UK conceptualized the study. CF, UK, LD and SE
performed the experiments and analysed the data. LP modelled the experimental data.
CF and UK wrote the manuscript. DBD provided the funding. All authors discussed the
findings, developed the analysis and generated the manuscript.

Shape evolution of pumice during granular flow

Carolina Figueiredo^{1*}, Ulrich Kueppers¹, Luiz Pereira^{1,2}, Lisa Depauli¹,
Sarp Esenyel¹, Donald B. Dingwell^{1,2}

¹ Department of Earth and Environmental Sciences, Ludwig-Maximilians-
Universität München, 80333 Munich, Germany.

² GEOLAB, Hangzhou International Innovation Institute, Beihang University,
Hangzhou, China

ORCID (CF): 0000-0003-1162-0240

ORCID (UK): 0000-0003-2815-1444

ORCID (LP): 0000-0001-9555-0352

ORCID (DBD): 0000-0002-3332-789X

*Corresponding authors: figueiredo.ca@lmu.de

Keywords: pyroclasts, pyroclastic density current, abrasion, explosive eruption,
relaxation theory.

**Abstract**

Explosive volcanic eruptions are a major geo-hazard whose lethality increases with
human population. Given the energetic nature of eruptive processes, direct
observation of key phenomena is limited. Accordingly, the study of volcanic deposits
and pyroclast textures is essential for understanding eruption and transport processes.
Experimental constraints provide a vital contribution to improving hazard assessment.

We performed three types of tumbling experiments using lapilli-sized phonolitic pumice
from the 13ka Laacher See eruption (Eifel, Germany) to investigate volcanic ash
generation and pyroclast shape evolution. Before and after **each experimental step**,
samples were sieved and one physical (volume) as well as four morphological
parameters (axial ratio, convexity, form factor, solidity) of 100 clasts were measured.
Most clast shape change happened in the first **15 minutes** and produced up to 50 wt.%
ash.

We frame our analysis in terms of effective relaxation timescales, whereby pyroclasts
display a decelerating rate of shape change towards a time-invariant shape (i.e.,
relaxation towards an equilibrium shape). This quantification of the susceptibility of
porous pyroclasts to shape and size changes enhances our understanding of transport
processes from clast generation to sedimentation. The related empirical law will allow
volcanologists to assess the shape- and time-dependent response to mechanical
exposure during transport.

**1. Introduction**

[revised manuscript text omitted]

*Geophysical Research: Solid Earth* **87**, 7061-7072 (1982).
<https://doi.org/10.1029/JB087iB08p07061>
- Pyle, D. M. The thickness, volume and grainsize of tephra fall deposits. *Bulletin of*
*Volcanology* **51**, 1-15 (1989). <https://doi.org/10.1007/bf01086757>
- Sparks, R. S. J., Bursik, M. I., Ablay, G. J., Thomas, R. M. E. & Carey, S. N. Sedimentation
of tephra by volcanic plumes. Part 2: controls on thickness and grain-size variations of
tephra fall deposits. *Bulletin of Volcanology* **54**, 685-695 (1992).
<https://doi.org/10.1007/bf00430779>
- Wilson, L. & Huang, T. C. The influence of shape on the atmospheric settling velocity
of volcanic ash particles. *Earth and Planetary Science Letters* **44**, 311-324 (1979).
[https://doi.org/10.1016/0012-821x\(79\)90179-1](https://doi.org/10.1016/0012-821x(79)90179-1)
- Dufek, J. & Bergantz, G. W. Suspended load and bed-load transport of particle-laden
gravity currents: the role of particle–bed interaction. *Theoretical and Computational*
*Fluid Dynamics* **21**, 119-145 (2007). <https://doi.org/10.1007/s00162-007-0041-6>
- Jones, T. J., Shetty, A., Chalk, C., Dufek, J. & Gonnermann, H. M. Identifying rheological
regimes within pyroclastic density currents. *Nature Communications* **15**, 4401 (2024).
<https://doi.org/10.1038/s41467-024-48612-7>
- Taddeucci, J. & Palladino, D. Particle size-density relationships in pyroclastic deposits:
inferences for emplacement processes. *Bulletin of Volcanology* **64**, 273-284 (2002).
<https://doi.org/10.1007/s00445-002-0205-6>
- Valentine, G. A. Stratified flow in pyroclastic surges. *Bulletin of Volcanology* **49**, 616-
630 (1987). <https://doi.org/10.1007/bf01079967>
- Buckland, H. M., Eychenne, J., Rust, A. C. & Cashman, K. V. Relating the physical
properties of volcanic rocks to the characteristics of ash generated by experimental
abrasion. *Journal of Volcanology and Geothermal Research* **349**, 335-350 (2018).
<https://doi.org/10.1016/j.jvolgeores.2017.11.017>
- Dufek, J. & Manga, M. In situ production of ash in pyroclastic flows. *Journal of*
*Geophysical Research: Solid Earth* **113** (2008). <https://doi.org/10.1029/2007jb005555>
- Walker, G. P. L. Generation and dispersal of fine ash and dust by volcanic eruptions.
*Journal of Volcanology and Geothermal Research* **11**, 81-92 (1981).
[https://doi.org/10.1016/0377-0273\(81\)90077-9](https://doi.org/10.1016/0377-0273(81)90077-9)
- Cagnoli, B. & Manga, M. Granular mass flows and Coulomb's friction in shear cell
experiments: Implications for geophysical flows. *Journal of Geophysical Research:*
*Earth Surface* **109** (2004). <https://doi.org/10.1029/2004if000177>
- Hornby, A., Kueppers, U., Maurer, B., Poetsch, C. & Dingwell, D. Experimental
constraints on volcanic ash generation and clast morphometrics in pyroclastic density
currents and granular flows. *Volcanica* **3**, 263-283 (2020).
<https://doi.org/10.30909/vol.03.02.263283>

- 19 Kueppers, U., Putz, C., Spieler, O. & Dingwell, D. B. Abrasion in pyroclastic density
currents: Insights from tumbling experiments. *Physics and Chemistry of the Earth, Parts*
*A/B/C* **45-46**, 33-39 (2012). <https://doi.org/10.1016/j.pce.2011.09.002>
- 20 Manga, M., Patel, A. & Dufek, J. Rounding of pumice clasts during transport: field
measurements and laboratory studies. *Bulletin of Volcanology* **73**, 321-333 (2011).
<https://doi.org/10.1007/s00445-010-0411-6>
- 21 Mueller, S. B., Lane, S. J. & Kueppers, U. Lab-scale ash production by abrasion and
collision experiments of porous volcanic samples. *Journal of Volcanology and*
*Geothermal Research* **302**, 163-172 (2015).
<https://doi.org/10.1016/j.jvolgeores.2015.07.013>
- 22 Schwarzkopf, L. M., Spieler, O., Scheu, B. & Dingwell, D. B. Fall-experiments on Merapi
basaltic andesite and constraints on the generation of pyroclastic surges. *eEarth* **2**, 1-
5 (2007). <https://doi.org/10.5194/ee-2-1-2007>
- Caballero, L., Sarocchi, D., Borselli, L. & Cárdenas, A. I. Particle interaction inside debris
flows: Evidence through experimental data and quantitative clast shape analysis.
*Journal of Volcanology and Geothermal Research* **231-232**, 12-23 (2012).
<https://doi.org/10.1016/j.jvolgeores.2012.04.007>
- Yao, T., Yang, H., Lourenço, S. D. N., Baudet, B. A. & Kwok, F. C. Y. Multi-scale particle
morphology evolution in rotating drum tests: Role of particle shape and pore fluid.
*Engineering Geology* **303** (2022). <https://doi.org/10.1016/j.enggeo.2022.106669>
- Lewin, J. & Brewer, P. A. Laboratory simulation of clast abrasion. *Earth Surface*
*Processes and Landforms* **27**, 145-164 (2002). <https://doi.org/10.1002/esp.306>
- Deal, E. *et al.* Grain shape effects in bed load sediment transport. *Nature* **613**, 298-302
(2023). <https://doi.org/10.1038/s41586-022-05564-6>
- Kragelsky, I. V. in *Friction and Wear* (eds I.V. Kragelsky, M.N. Dobychin, & V.S.
Kombatov) 297–316 (Pergamon, 1982).
- Campbell, I. A. & Giovannella, C. Relaxation in Complex Systems and Related Topics.
*Springer* (1990). <https://doi.org/10.1007/978-1-4899-2136-9>
- Jaeger, H. M. Celebrating Soft Matter’s 10th Anniversary: Toward jamming by design.
*Soft Matter* **11**, 12-27 (2015). <https://doi.org/10.1039/c4sm01923g>
- Kohlrausch, R. Theorie des elektrischen Rückstandes in der Leidener Flasche. *Annalen*
*der Physik* **167**, 179-214 (1854). <https://doi.org/10.1002/andp.18541670203>
- Williams, G. & Watts, D. C. Non-symmetrical dielectric relaxation behaviour arising
from a simple empirical decay function. *Transactions of the Faraday Society* **66** (1970).
<https://doi.org/10.1039/tf9706600080>
- Morales Rivera, A. M., Amelung, F., Albino, F. & Gregg, P. M. Impact of Crustal Rheology
on Temperature-Dependent Viscoelastic Models of Volcano Deformation: Application
to Taal Volcano, Philippines. *Journal of Geophysical Research: Solid Earth* **124**, 978-994
(2019). <https://doi.org/10.1029/2018jb016054>
- Lancelotti, R. F., Zanotto, E. D. & Sen, S. Kinetics of physical aging of a silicate glass
following temperature up- and down-jumps. *The Journal of Chemical Physics* **160**
(2024). <https://doi.org/10.1063/5.0185538>
- Lancelotti, R. F., Pereira, L., Hess, K.-U., Dingwell, D. B. & Zanotto, E. D. Flash-DSC
provides valuable insights into glass relaxation and crystallization. *Journal of Non-*
*Crystalline Solids* **646** (2024). <https://doi.org/10.1016/j.jnoncrysol.2024.123242>
- Schmincke, H.-U., Park, C. & Harms, E. Evolution and environmental impacts of the
eruption of Laacher See Volcano (Germany) 12,900 a BP. *Quaternary International* **61**,
61-72 (1999). [https://doi.org/10.1016/s1040-6182\(99\)00017-8](https://doi.org/10.1016/s1040-6182(99)00017-8)

- Cleary, P. W. DEM simulation of industrial particle flows: case studies of dragline
excavators, mixing in tumblers and centrifugal mills. *Powder Technology* **109**, 83-104
(2000). [https://doi.org/10.1016/s0032-5910\(99\)00229-6](https://doi.org/10.1016/s0032-5910(99)00229-6)
- Freundt, A. & Schmincke, H. U. Abrasion in pyroclastic flows. *Geologische Rundschau*
**81**, 383-389 (1992). <https://doi.org/10.1007/bf01828605>
- Breard, E. C. P. *et al.* The fragmentation-induced fluidisation of pyroclastic density
currents. *Nature Communications* **14** (2023). <https://doi.org/10.1038/s41467-023-37867-1>
Cigala, V. *et al.* The dynamics of volcanic jets: Temporal evolution of particles exit
velocity from shock-tube experiments. *Journal of Geophysical Research: Solid Earth*
**122**, 6031-6045 (2017). <https://doi.org/10.1002/2017jb014149>
- Radjai, F., Roux, J.-N. & Daouadji, A. Modeling Granular Materials: Century-Long
Research across Scales. *Journal of Engineering Mechanics* **143** (2017).
[https://doi.org/10.1061/\(asce\)em.1943-7889.0001196](https://doi.org/10.1061/(asce)em.1943-7889.0001196)
- Yu, F. *et al.* Particle breakage of sand subjected to friction and collision in drum tests.
*Journal of Rock Mechanics and Geotechnical Engineering* **13**, 390-400 (2021).
<https://doi.org/10.1016/j.jrmge.2020.08.004>
- Kirchner, N. P. Thermodynamically consistent modelling of abrasive granular materials.
I Non-equilibrium theory. *Proceedings of the Royal Society of London. Series A:*
*Mathematical, Physical and Engineering Sciences* **458**, 2153-2176 (2002).
<https://doi.org/10.1098/rspa.2002.0963>
- Zorn, E. U. *et al.* Experimental investigation of volcanoclastic compaction during burial.
*Volcanica* **7**, 765-783 (2024). <https://doi.org/10.30909/vol.07.02.765783>
- Calder, E. S. *et al.* Mobility of pyroclastic flows and surges at the Soufriere Hills Volcano,
Montserrat. *Geophysical Research Letters* **26**, 537-540 (1999).
<https://doi.org/10.1029/1999gl900051>
- Calder, E. S., Sparks, R. S. J. & Gardeweg, M. C. Erosion, transport and segregation of
pumice and lithic clasts in pyroclastic flows inferred from ignimbrite at Lascar Volcano,
Chile. *Journal of Volcanology and Geothermal Research* **104**, 201-235 (2000).
[https://doi.org/10.1016/s0377-0273\(00\)00207-9](https://doi.org/10.1016/s0377-0273(00)00207-9)
- Houghton, B. F., Wilson, C. J. N., Fierstein, J. & Hildreth, W. Complex proximal
deposition during the Plinian eruptions of 1912 at Novarupta, Alaska. *Bulletin of*
*Volcanology* **66**, 95-133 (2004). <https://doi.org/10.1007/s00445-003-0297-7>
- Branney, M. J. & Kokelaar, B. P. *Pyroclastic Density Currents and the Sedimentation of*
*Ignimbrites*. Vol. 27 (Geological Society, 2002).
- Daniel, R. C., Poloski, A. P. & Eduardo Sáez, A. A continuum constitutive model for
cohesionless granular flows. *Chemical Engineering Science* **62**, 1343-1350 (2007).
<https://doi.org/10.1016/j.ces.2006.11.035>
- Fang, C., Wang, Y. & Hutter, K. A thermo-mechanical continuum theory with internal
length for cohesionless granular materials. *Continuum Mechanics and*
*Thermodynamics* **17**, 545-576 (2006). <https://doi.org/10.1007/s00161-006-0007-8>
- Fang, C., Wang, Y. & Hutter, K. Shearing flows of a dry granular material—hypoplastic
constitutive theory and numerical simulations. *International Journal for Numerical and*
*Analytical Methods in Geomechanics* **30**, 1409-1437 (2006).
<https://doi.org/10.1002/nag.525>
- Fang, C. & Wu, W. On the weak turbulent motions of an isothermal dry granular dense
flow with incompressible grains: part I. Equilibrium turbulent closure models. *Acta*
*Geotechnica* **9**, 725-737 (2014). <https://doi.org/10.1007/s11440-014-0313-4>

- Göncü, F. & Luding, S. Effect of particle friction and polydispersity on the macroscopic
stress–strain relations of granular materials. *Acta Geotechnica* **8**, 629–643 (2013).
<https://doi.org/10.1007/s11440-013-0258-z>
- Volfson, D., Tsimring, L. S. & Aranson, I. S. Partially fluidized shear granular flows:
continuum theory and molecular dynamics simulations. *Phys Rev E Stat Nonlin Soft*
*Matter Phys* **68**, 021301 (2003). <https://doi.org/10.1103/PhysRevE.68.021301>
- Bagdassarov, N. S. & Dingwell, D. B. A rheological investigation of vesicular rhyolite.
*Journal of Volcanology and Geothermal Research* **50**, 307–322 (1992).
[https://doi.org/10.1016/0377-0273\(92\)90099-y](https://doi.org/10.1016/0377-0273(92)90099-y)
- Bagdassarov, N. S. & Dingwell, D. B. Frequency dependent rheology of vesicular
rhyolite. *Journal of Geophysical Research: Solid Earth* **98**, 6477–6487 (1993).
<https://doi.org/https://doi.org/10.1029/92JB02690>
- Bagdassarov, N. S. & Dingwell, D. B. Deformation of foamed rhyolites under internal
and external stresses: an experimental investigation. *Bulletin of Volcanology* **55**, 147–
154 (1993). <https://doi.org/10.1007/bf00301512>
- Cormier, L., Galois, L., Lelong, G. & Calas, G. From nanoscale heterogeneities to
nanolites: cation clustering in glasses. *Comptes Rendus. Physique* **24**, 199–214 (2023).
<https://doi.org/10.5802/crphys.150>
- Greaves, G. N. EXAFS and the structure of glass. *Journal of Non-Crystalline Solids* **71**,
203–217 (1985). [https://doi.org/10.1016/0022-3093\(85\)90289-3](https://doi.org/10.1016/0022-3093(85)90289-3)
- Le Losq, C. *et al.* Percolation channels: a universal idea to describe the atomic structure
and dynamics of glasses and melts. *Scientific Reports* **7** (2017).
<https://doi.org/10.1038/s41598-017-16741-3>
- Lancelotti, R. F. *et al.* Physical aging of lithium disilicate glass. *Journal of Non-Crystalline*
*Solids* **622** (2023). <https://doi.org/10.1016/j.jnoncrysol.2023.122661>
- Lancelotti, R. F., Rodrigues, A. C. M. & Zanutto, E. D. Structural relaxation dynamics of
a silicate glass probed by refractive index and ionic conductivity. *Journal of the*
*American Ceramic Society* **106**, 5814–5821 (2023). <https://doi.org/10.1111/jace.19285>
- Pisello, A. *et al.* The porosity of felsic pyroclasts: laboratory validation of field-based
approaches. *Bulletin of Volcanology* **85**, 69 (2023). <https://doi.org/10.1007/s00445-023-01679-4>
- Liu, E. J., Cashman, K. V. & Rust, A. C. Optimising shape analysis to quantify volcanic
ash morphology. *GeoResJ* **8**, 14–30 (2015). <https://doi.org/10.1016/j.grj.2015.09.001>
- Figueiredo, C. A., Bongiolo, E. M., Jutzeler, M., da Fonseca Martins Gomes, O. &
Neumann, R. Alkaline pyroclast morphology informs on fragmentation mechanisms,
Trindade Island, Brazil. *Journal of Volcanology and Geothermal Research* **428** (2022).
<https://doi.org/10.1016/j.jvolgeores.2022.107575>

**Author contributions.** CF and UK conceptualized the study. CF, UK, LD and SE
performed the experiments and analysed the data. LP analysed the experimental data.
CF and UK wrote the manuscript. DBD provided the funding. All authors discussed the
findings, developed the analysis and generated the manuscript.

Reviewer #1:

1. *The authors show that, under their experimental conditions, shape of pyroclasts rapidly evolves due to abrasion towards a "final" state. In other words, the relaxation times tend to be short. From this, the authors conclude, among other observations, that abrasion in the conduit is probably small considering the irregular, angular shape shown by lapilli clasts in fallout deposits. I understand the logic behind this, but there are relevant processes, especially shortly after fragmentation, that may generate irregular morphologies such as clasts breaking and mask the morphological effects of abrasion due to the continuous generation of new irregular clasts. I think experiments may suggest that abrasion is small in the plume, but it is not clear to me this conclusion in the conduit. I suggest to discuss this topic.*

Response

Thank you for the comment and suggestion. We stress however that the main outcome of our manuscript was not about conditions during conduit flow. We concur that collisions may take place from above the fragmentation level all the way to sedimentation. In our experiments, clast velocity and contact energy are small, yet clast rounding is happening readily. Our statement regarding the low probability of clast interactions during conduit flow referred to the related effects on clast shape, not clast size. The main findings of our manuscript are that 1) ash-sized clasts are generated readily, 2) the amount of ash generated correlates to experimental conditions and 3) the shape evolution correlates to experimental duration. The implications about the dynamics of gas-particle flow in the conduit and the eruption column are just one aspect of the discussion in the manuscript.

We use two lines of evidence to believe that particle-particle and particle-wall/substrate interactions are of minor importance regarding the pyroclasts (coarse ash and larger) in fall deposits. We say this first because direct observations of high-speed videos of laboratory experiments (Cigala et al., 2017) and Strombolian eruptions (Taddeucci et al., 2012) indicate a dilatational setting at fragmentation level where the distance between clasts dominantly increases and a strong nonlinear decay of ejection velocity with time was observed. Collisions take place but of rather low intensity because clasts are moving in the same direction. Fully collisional interactions may indeed lead to secondary fragmentation if the differential velocity is high enough. Tangential collisions would alter the shape of one (or both) clasts by removing surface asperities, causing a rounded surface texture of clasts. Subsequent fully collisional interactions would then lead to clasts exhibiting one (or more) fracture planes while part of the surface would show signs of surface roughness overprinting, i.e. rounding (Kueppers et al., 2013, AGU poster). We do not appear to have observed this in natural deposits.

Our experiments were intentionally designed to be a simplified and low-energy yet well-controlled and repeatable process and that processes in nature are dynamic and complex. Yet, the fast shape evolution even in these experiments

led us to the conclusion that angular pyroclasts in fall deposits prove minor to no surface roughness overprinting upon clast formation. We have adjusted the text accordingly and furthered the discussion on the topic (lines 309-353).

2. *How comparable are the experiments with granular flows that propagate with higher temperatures? This likely changes the mechanical properties of clasts, which are critical to define the efficiency of abrasion and clasts breaking.*

Response

Thank you for the comment. Our experiments are designed to quantitatively map shape alteration (and related ash generation) of porous clasts under controlled and repeatable conditions. These experiments are taking place at largely constant conditions and do therefore not account for the dynamic variations in nature. In fact, laboratory experiments should not aim at this as otherwise it remains almost impossible to derive empirical laws about the influence of a single parameter.

The effect of high temperatures on the mechanical properties of pyroclasts during PDCs remains unexplored. It is known that material properties like Young's modulus and tensile strength are temperature-dependent, but the extent of these changes is influenced not only by composition but also by factors such as the amount of phenocrysts and overall rock texture. Scheu et al. (2012, AGU poster) compared experimentally generated clasts in decompression experiments at room temperature and 850 °C and found that there is a negative correlation of fragmentation efficiency with sample temperature. In uniaxial compression texts, studies by Lamur et al. (2018), Benson et al. (2012), and Rocchi et al. (2004) have investigated these properties at temperatures up to 1000°C, showing that the mechanical response varies significantly with texture. Given the low energy of our experiments that cannot be performed at >100°C, we do not speculate about the effect of bulk gravity current temperature on abrasion efficiency. We do however expect that hotter currents that are capable of entraining and heating more ambient air will also be subject to higher degrees of ash depletion by elutriation. To fully understand the impact of temperature on abrasion and clast breakage during granular flows, dedicated high-temperature experiments would be required. This is outside the scope of this study.

3. *The authors use steel balls to enhance abrasion (as done by lithics for instance) and, as expected, they have a relevant effect in experimental results. Again, it is not clear the natural conditions, if any, for which these experimental conditions are relevant.*

Response

We thank the reviewer for this comment. Many explosive eruptions expel clasts that do not derive from the magma body feeding a specific eruption. These clasts are called lithics and are commonly of lower bulk density compared than those deriving from the magma driving an eruption. In our experiments, steel balls were used as a proxy to represent the presence of lithic clasts in pyroclastic density currents. We acknowledge that this is a simplification. We agree that steel does not mimic the full physical properties of natural lithics, and we have clarified this limitation in the revised manuscript (Lines 488–491). Our intention is not to replicate a specific natural setting, but rather to demonstrate the potential influence and magnitude of lithic clasts on pumice degradation under conditions

of enhanced abrasion. Simultaneous transport of juvenile clasts and lithics in PDCs does not necessarily imply that all clasts move at the same speed. We used steel balls (in contrast to Hornby et al. 2020) as non erodible “clasts” to allow for the attribution of “all” ash to originate from the pumice.

4. *I think Figure 2 can be modified in order to be clearer. Titles above SM are confusing. I suggest to present the labels "Convexity", "Form Factor" and "Solidity" in the y-axis, and a label of SM above the first box in each panel.*

Response

We thank for these suggestions and have changed the figure in the revised manuscript.

5. *If the allowed number of figures limit allows it, I suggest to extend the methodology by including a figure with a scheme of the experimental device. On the other hand, the information of specific clasts in Figure 4 (I mean, the shadowgraphs) is too small and could be improved. Finally, in Figure 5, I think the variable τ should be expressed in time dimensions.*

Response

We thank the reviewer for these helpful suggestions. A schematic of the experimental setup is already available in Hornby et al. (2020), which we now cite explicitly in the Methods section, so we believe it is not necessary to include an additional figure here. In response to the comment on Figure 4, we have enlarged the shadowgraph images and adjusted the layout to improve visibility and better convey the clast shape evolution. Additionally, we have revised Figure 5 to express τ in time units (minutes).

6. *In summary, even if experiments are useful tools to evaluate the shape evolution dynamics of pyroclasts in granular flows, I think that results are maybe overinterpreted in terms of the analysis of natural volcanic phenomena because it is not clear under which conditions natural systems can be described by means of this experimental configuration. The authors mention that granular flows present physical characteristics for which we could expect an even higher abrasion/breaking capacity, but this should be discussed in detail. What is the solid fraction in experiments? Do we have an idea of energy involved in collisions? What kind of granular flow (or part of a granular flow) it is expected to represent?*

Response

Thank you for the comment and insightful questions. We have added an explanation in the discussion (lines 329-334) clarifying the type of natural deposits that our experiments aim to represent. We compare our experiments specifically with granular flows and the dense underflow portions of pyroclastic density currents, where clast-supported transport and particle-particle and particle-substrate interactions are dominant. We repeat that those experiments serve to map out the influence of boundary conditions on clast abrasion and associated ash generation. We stress that those experiments are at the lower energy range that can – at times – be reached in nature but that we guarantee stable conditions and not dynamic conditions as in nature. Additionally, we state

that we have a high granular temperature = clasts (quasi) always in contact. Accordingly, dynamic conditions in nature where lower particles concentrations (= clasts not always in contact) but potentially higher differential velocities (= higher collisional energy) may compensate for the experimental conditions. However, our experiments do not intend to quantify the degree of abrasion of pyroclasts during PDCs but serve to highlight the speed at which initially angular clasts were abraded and rounded, which has been also added to the discussion (lines 233-237). The empirically derived speed serves to constrain where, when and/or how efficiently particle-particle and/or particle-substrate interaction has taken place when analysing natural deposits.

Reviewer #2:

1. *The major concern I have with this study and analysis is why weren't the grains mapped or characterised in 3D? The methods exist from simple 3D scanning techniques through to SEM, MCT, and even synchrotron analysis. The 2D characterisation of already fragmented and deposited grains will have a range of surface features/textures, porosity, and possibly crystal features that are not well represented in the measures undertaken, which may have some influence on the outcome. Surely, density, porosity, and some measure of hardness, glass surface structures would also be interesting to explore alongside grain surface textures/shapes than just analysing grain shape parameters. In my opinion, expanding the measures of the properties of the grains would advance this study.*

Response

We thank the reviewer for this valuable comment. We agree that 3D grain characterization is capable of providing a comprehensive description of clast morphology, yet they are time-consuming and costly. 2D analysis as presented here is much faster and was intentionally developed. It will allow for deriving a material property (= shape based on image) taken in the field to avoid any shape changes after sampling during transport all the way to the lab (as we have proven that shape changes occur very rapidly....). For our analysis, we assume that the shape roughness characteristics are not orientation-dependent. We have amended the text in lines 510-513. We agree that integrating additional physical properties and textural characterization would enrich the study, and we intend to explore these aspects in future work.

2. *The sample set of pumice lapilli is, in my opinion, too limited to be able to make broad conclusions about pyroclastic flow processes. I would expect that a range of pumice material was tested based on a range of pumice properties to see if the results are a standard response to the tumbling. I would expect that just using pumice from one unit within an eruption, where the material has already been fragmented and transported, may skew the results. While I understand that it is probably impossible to test newly fragmented, erupted lapilli before it has transformed into a flow, there must be some proximal pumice units worldwide that represent the erupted material of an eruption column that is coeval to a pyroclastic flow that would be a better target to sample.*

Response

Thank you for the comment. We acknowledge that the data presented in this manuscript using pumice from a single unit are not universally applicable. The textural influence has already been shown to some degree by Hornby et al. (2020). We will use different lithologies for comparative studies but believe that the presented results merit publication at this point as it will help to better understand clast interaction during explosive eruptions. The material used in this study comes from a well-characterized fallout deposit of the Laacher See eruption. We carefully sampled pristine fall units for lapilli-sized clasts as starting material. We elaborated the text in lines 95-99 and 456-460.

Although our experiments allow for quantitative description of ash generation as a function of time, it is important to stress that the main finding is the fast propensity of porous lapilli to change shape in all experiments. Time and experiment time clearly control ash generation and shape change. Ash

generation and shape evolution are decreasing in their efficiency with time and allow direct conclusions about where and when particle-particle and particle-wall/substrate interactions were likely or important enough to change particle shape (not size).

3. *As the experiment is meant to simulate ash generation in a pyroclastic flow, I would expect the discussion to have some more detail around where this process would occur in the flow (spatially and temporally) and how the fines contribute to the flowing mass and whether any sedimentological descriptions of real-world flow deposits show an increase in fines either laterally/longitudinally or vertically, however I understand the need to limit the amount of text in this format. I also wonder if the relationships found in Figure 5 can be related to the real-world flows/deposits or even experimental flows. This may be beyond the scope of this study, but I am sure if the research progresses, it would be an important comparison.*

Response

Thank you for the comment. As previously mentioned, we have added clarification in the discussion (lines 329-333). We stress that we are investigating transport processes here and not stratigraphical details of sediments. We showed with set B experiments that ash generation is a very fast process (answering the comment “temporally”). “Spatially”, the sample load is in constant movement. Comparing T1 and T2 experiments shows clearly how the presence of ash is reducing the collisional energy and leading to reduced ash generation efficiency in T1 experiments. Moreover, the constant generation of volcanic ash in the lower part of PDCs (assuming that most interaction is taking place here) will surely affect the mobility of PDCs. Constant replenishment of the ash-rich medium will keep permeability low and accordingly act against PDC condensation, promoting a longer run-out distance. How the fines contribute to the flowing mass has been discussed in lines 191-218 of the discussion. We have revised and strengthened the discussion regarding the applicability of the relaxation theory to pyroclastic density currents. We believe that the presented model has the potential to set the base for future more complete data sets. Similar experiments complementing the data presented here are planned and will increase the robustness of the empirical findings acquired.

4. *I am not sure of the relevance of further investigating the grain size of the newly produced ash and whether there needs to be further presentation and analysis of grain sizes below 2mm. In some flows, fines or clay-sized particles may contribute to changing flow conditions. I understand that moving beyond tumbling in a concrete mixer is most likely beyond this study, but characterising the volumes of grain sizes below 2 mm may inform future experimental analysis on flow behaviours.*

Response

Thank you for the comment. We have conducted grain size distribution analyses of the ash produced during the experiments and found that, while the absolute amounts vary, the grain size distribution curves are essentially identical across different experimental conditions. These findings are the outcome of a soon-to-be submitted thesis and will be published separately.

5. *I would also recommend removing any reference or text (lines ~250-290) that tries to relate these experiments to conduit processes or fragmentation, as I do not see how these experiments, tumbling of lapilli in a cement mixer, relate to or scale to the conduit fragmentation process. While I understand that ash is produced in those processes, the experimental design is probably more aligned to collision and (low energy) frictional processes in a flow. Focusing on pyroclastic flow processes also presents a more focused study and manuscript.*

Response

Thank you for this comment. As laid out earlier, our experiments surely do not mimic directly the transport conditions of gas-particle jets in conduits. Yet, we believe that our experiments can convincingly show that low-energy tumbling (e.g. small bed load, slow transport velocity, room T) is capable of strong shape alteration of initially angular porous particles (as expected for explosive eruptions involving intermediate to felsic magmas). We are taking the findings of the experiments presented here to map out areas of pyroclast transport (from fragmentation to sedimentation and compaction) where shape alterations may be important enough to be constrained by comparative studies of particle shape.

6. *A minor change I could consider would be changing the heading at 4.2 to morphological properties, not petrophysical properties, as only shape properties have been determined; no lithological properties have been determined, such as porosity, etc.*

Response

Thank you for the suggestion. We agree that "morphological properties" is a more accurate heading, as our analyses focus solely on shape-related parameters (shape analysis and volume) rather than broader petrophysical properties. We changed accordingly.

7. *The results and analysis of the manuscript support the conclusions. In general, the discussion could be more focused on pyroclastic flow processes, yet overall, the study does present some interesting starting points for future (postgraduate student) research. The manuscript will influence some further investigation into future thinking in the field and most likely influence the analysis of large-scale experimental flows and how they are analysed during flow, as well as deposits are analysed.*

Response

Thank you for your positive feedback. We are glad that the results and conclusions are seen as well supported and appreciate your recognition of the study's potential to inform future research. Following your suggestion, we have refined the discussion to better focus on pyroclastic flow processes and have more clearly defined the natural conditions our experiments aim to represent.

8. *The manuscript could be improved through recognising the limitations of the experiments, in particular the sample material gathered, and the lack of repeatability or the number of experiments conducted. Stronger justification of the clast shape parameters chosen could be provided, and why these are*

characteristics best related to the physical process of ash generation, and how other lithological parameters may provide a more holistic view of factors contributing to ash generation.

Response

We thank the reviewer for the comment but disagree in the judgement of experiments. We displayed three different sets of experiments to ensure repeatability, and it lies in the nature of experiments to reduce size and complexity of natural systems to allow for mapping the influence of individual parameters under controlled and repeatable conditions. However, we agree on the importance of recognising the limitations of the experimental setup and therefore we have added a paragraph on lines 456–460 that addresses this, including the sample material used and the number of experiments conducted, and outlined the need for future analyses with different materials and the integration with other lithological parameters. We have also added an explanation for the choice of shape parameters in the methods section (lines 508–510).

Reviewer #3:

1. *It is necessary to add a table with main experimental conditions (T1, T2, T3b) and results (for example, amount of ash produced) at the beginning of the results chapter to help readers follow data presentation.*

Response

Thank you for the comment. The main experimental conditions for T1, T2, and T3B are already detailed in the Methods section, and the amount of ash produced is presented in Figure 1. To improve clarity, we have added a reference to the methods section at the beginning of the Results chapter.

2. *For better comparison, it is necessary that solidity, convexity and form factor axes in figures 2 and 3 display the same intervals and size.*

Response

We thank the reviewer for this observation. We chose to display different y-axis intervals in Figures 2 and 3 to optimize the visualization of each parameter's variability; the axes are scaled individually to best reflect the range and distribution of the data within each plot.

3. *Morphology: Convexity, form factor and solidity do not significantly (as seen from boxplots) change after the first 15 min of tumbling and in different experimental conditions in fig. 2 experiments but they seem to evolve in fig. 3 experiments. Why is that? where pumices different? did the coloring affect their mechanical properties? in the lines 364-374 authors suggest that their textures could be different; if so, the trend should not be considered of general value.*

Response

We appreciate the close inspection of our figures. Figure 2 presents data from set A experiments where 100 random clasts from the experimental load (2 kg) were investigated for their shape evolution after 15, 30, 45, 60, and 120 minutes increments, respectively. Figure 3 presents the results of set B experiment T1 where shorter time increments were applied (5, 10, 15, 20, and 60 minutes, respectively). Here, 100 clasts dyed with food colour were part of the experimental load (2 kg) and could be identified after each time increment. In essence, figure 3 refers to Set B, which includes shorter time steps, allowing us to capture the faster shape change compared to set A experiments.

We state that the colouring process (aqueous solution, no epoxy, no lacquer) did not affect their mechanical properties; weight measurements before and after colouring showed a change of less than 1%, which is within measurement uncertainty. Therefore, the observed differences are due to improved temporal resolution and clast tracking, not the colouring process.

4. *Implications are: in a pdc deposit despite continuous ash production in the current lapilli morphology does not change/change from medial to distal locations. Please check for deposit data confirming this- what about the general shapes of clasts in real pdc/fall deposits?*

Response

Thank you for your comment. We have added a reference to Manga et al. (2011) in lines 343–344, highlighting that morphology changes occur quite readily. Regarding the overall shape of clasts in natural PDCs and fall deposits, we mention in the Introduction (lines 53) that clasts from fall deposits typically display an angular shape. Additionally, we have now added clarification in line 60-61, stating that clasts in PDCs tend to exhibit a more rounded morphology. Transport conditions in natural PDCs are highly dynamic, so will ash generation be. However, particle-particle interaction given, there is consensus that mechanically weak clasts in PDCs will become rounder with distance and hence loose volume/weight. Deposits, both vertically in one outcrop but also at different distance from the vent will show textural differences that reflect changes in time and space. Our experiments happen in a closed system where no sorting or preferential enrichment/depletion takes place.

5. *in the discussion, the authors compare results with granular flow- I see the effect of gas escape and ash elutriation as way less relevant than other concentrated (i.e. fluidised currents). Could they provide some example of observed deposits (for example, in recent eruptions of Lascar volcano).*

Response

Thank you for your comment and suggestion. We have added a paragraph in the discussion that provides a more specific comparison between our experiments and natural deposits and added a couple of examples, including Lascar Volcano deposits (lines 329-334). We strengthened the discussion to avoid conceptual misunderstanding between the meaningfulness of the experiments without their direct applicability to decipher sedimentological features in stratigraphic sections.

6. *Fig. 4 and associated data are still part of the results obtained in the study and should be presented as such, not in the discussion chapter.*

Response

Thank you for the suggestion. We agree, Fig. 4 and the associated data have been moved from the Discussion to the Results section in the revised manuscript.

7. *While three orthogonal axes of the lapilli were measured for volume calculation, no 3d analysis was made for shape. This is a weakness of this study. I believe the authors have data to show shape variations on 3d Sneed and Folk classification plots. These plots should be added to the article.*

Response

Thank you for the suggestion. We have plotted the data in the 3D Sneed and Folk (1978) classification diagram and found that it does not contribute meaningfully to the interpretation of our results. This diagram categorizes fragments into broad shape classes (cubic, elongate, platy, and bladed), which do not reflect the finer-scale rounding and surface evolution observed during tumbling. Moreover, as shown in the manuscript, the axial ratios remain largely constant, resulting in no visible trend in the suggested plot. For these reasons, we decided not to include it in the paper.

Shape evolution of pumice during granular flow

Carolina Figueiredo^{1*}, Ulrich Kueppers¹, Lisa Depauli¹, Sarp Esenye¹, Luiz Pereira¹, Donald B. Dingwell.¹

¹ Department of Earth and Environmental Sciences, Ludwig-Maximilians-Universität München, 80333 Munich, Germany.

⁶ ORCID (CF): 0000-0003-1162-0240

⁷ ORCID (UK): 0000-0003-2815-1444

⁸ ORCID (LP): 0000-0001-9555-0352

⁹ ORCID (DBD): 0000-0002-3332-789X

*Corresponding authors: figueiredo.ca@lmu.de

Keywords: pyroclasts, pyroclastic density current, abrasion, explosive eruption, relaxation theory.

Abstract

Explosive volcanic eruptions are a major geo-hazard whose lethality grows daily with human population. Given the energetic nature of eruptive processes, direct observation of key phenomena is limited or impossible. Accordingly, the study of volcanic deposits and pyroclast textures are essential for understanding eruption and transport processes. In parallel, experimental constraints increasingly provide a vital contribution to improving hazard assessment.

We performed three types of tumbling experiments using pumice lapilli from the 13ka Laacher See eruption (Eifel, Germany) to investigate volcanic ash generation and pyroclast shape evolution. To this end, samples were sieved and two petrophysical (volume, porosity) as well as four morphological parameters (Axial Ratio, Convexity, Form Factor, Solidity) of 500- μ m clasts were measured before start and after five experimental steps. We observed the strongest shape change in the first 15 minutes and up to 50wt.% of ash generation.

We frame our analysis in terms of effective relaxation timescales, whereby pyroclasts display a decelerating rate of change towards a time-invariant shape. This quantification of the susceptibility of porous pyroclasts to shape and size changes enhances our understanding of transport processes. The related empirical law will allow volcanologists to assess the shape- and time-dependent response to mechanical exposure during transport.

1. Introduction

During the explosive eruption of felsic magma, fragmentation is dominated by brittle failure^{1,2}. The efficiency of this process depends on textural characteristics (e.g., porosity, crystallinity, permeability) and the transient gas overpressure in pore space, resulting from the competition between degassing and outgassing¹⁰. Upon fragmentation, angular porous pyroclasts of variable grain size are accelerated and eventually ejected into the atmosphere^{3,4}. The eruptive and transport dynamics control

Summary of Comments on 4. Reviewer 1 answers.pdf

Page: 1

 Number: 1 Author: Date: 30.03.2025 06:35:43
and size?

Status

carolinaalmeida Rejected 05.05.2025 09:18:39

 Author: carolinaalmeida Subject: Sticky Note Date: 14.08.2025 17:00:31

Thanks for the suggestion, however we would like to focus on the morphology aspect in our study.

 Number: 2 Author: Subject: Texto insertado Date: 30.03.2025 06:35:52
improve

Status

carolinaalmeida Rejected 03.05.2025 15:07:24

 Author: carolinaalmeida Subject: Sticky Note Date: 03.05.2025 15:08:13

To improve is grammatically incorrect.

 Number: 3 Author: Subject: Texto insertado Date: 30.03.2025 06:36:06
lapilli-sized phonolitic pumice

Status

carolinaalmeida Accepted 03.05.2025 15:06:14

 Number: 4 Author: Date: 30.03.2025 06:36:17

Status

carolinaalmeida Accepted 03.05.2025 15:11:14

 Number: 5 Author: Date: 30.03.2025 06:38:59

Status

carolinaalmeida Accepted 03.05.2025 15:11:22

 Number: 6 Author: Date: 30.03.2025 06:39:12

Is the term "effective" needed?

 Author: carolinaalmeida Subject: Sticky Note Date: 03.05.2025 15:10:13

It is relevant because it means that it was observed.

 Number: 7 Author: Date: 30.03.2025 06:39:21
Basaltic magmas can also be fragmented brittlely. I know that you are not mentioning the opposite affirmation, but somebody could interpret this idea, and thus I suggest to rephrase.

 Author: carolinaalmeida Subject: Sticky Note Date: 03.05.2025 15:17:55

Thanks for the suggestion, however it is clear from the context that we are referring specifically to felsic magmas.

 Number: 8 Author: Date: 30.03.2025 06:39:29
typically

Status

carolinaalmeida Accepted 03.05.2025 15:14:28

 Number: 9 Author: Subject: Texto insertado Date: 30.03.2025 06:39:35

, magma ascent conditions,

Status

carolinaalmeida Accepted 03.05.2025 15:13:01

 Number: 10 Author: Date: 30.03.2025 06:39:45

I suggest to cite:

Cassidy, M., Manga, M., Cashman, K., & Bachmann, O. (2018). Controls on explosive-effusive volcanic eruption styles. Nature communications, 9(1), 2839.

Status

carolinaalmeida Accepted 03.05.2025 15:25:21

the final distribution of the pyroclastic deposits. Fall (gravitational settling dominated)
and flow (sub-horizontal transport) deposits are the two end member deposit types of
such explosive events. The meaning of the grain size distribution as revealed from
volcanic deposits is a matter of discussion as it may depend – at variable degrees - on
primary (magma fragmentation) and post-fragmentation transport processes.

Fall deposits tend to be characterised by the effects of transport-related sorting due to
atmospheric conditions (e.g., wind field, humidity) as well as particle density and
aerodynamic properties⁵⁻⁸. In medial to distal regions, such deposits commonly contain
predominantly angular lapilli-sized clasts and exhibit evidence of density- and/or grain
size-sorting. In contrast, pyroclastic density currents (PDCs) advance rapidly down
volcanic flanks, experiencing both density-stratification and turbulence. Particle
concentration and flow turbulence define a wide spectrum of these currents, ranging
from particle-poor (pyroclastic surges) to granular flows with high granular
temperature. In the latter, particle-particle and particle-substrate interactions are
anticipated to be ubiquitous⁹⁻¹². Porous pyroclasts accordingly undergo evolution in
size and shape¹³⁻¹⁵, associated with the generation of fine particles, including those
<2mm, termed volcanic ash. The ash content, either part of a flow from the beginning
or formed in-situ by collisional/frictional processes, affects the flow's mobility, enlarging
the areas at threat.

The efficiency and impact of the experimental conditions on such size and shape
evolution of pyroclasts has been investigated experimentally via (1) tumbling¹⁶⁻¹⁹ and
(2) drop^{20,21} experiments as well as with the aid of numerical models¹⁴. Clast abrasion
and morphology evolution have also been evaluated for gravel particles in dry and wet
settings, relevant to debris flows^{22,23} and bed load transport in rivers²⁴. Those studies
indicate a strong influence of experimental conditions on clast shape evolution and the
resulting dynamic coupling to the surrounding medium²⁵. Thus, in principle,
experiments are capable of accessing the variation of morphological parameters.
Independent of the experimental conditions and material properties, a widely
recognised principle of particle abrasion remains which states that if a particle does
not fracture anew during the process, then its tendency to change shape decreases
over time eventually reaching a time-invariant 'equilibrium state' in terms of shape and
roughness²⁶.

The physical concept of relaxation was initially proposed to describe the dynamics of
gases and applied to dielectric materials. Nowadays, it has been used across various
systems, including glass science, superconductors, jamming in granular materials as
well as biological and magnetic systems²⁷⁻³⁰. It is vastly used to study structural
relaxation of glasses, whereby a structure and consequently the material properties
evolve to equilibrium at a rate dictated by a characteristic relaxation time^{31,32}. During
relaxation, the rate of change of a property commonly decelerates as this property
approaches the equilibrium or optimal state³¹. Here we investigate the shape relaxation
of abrading lapilli using laboratory experiments.

Number: 1 Author: Date: 30.03.2025 20:22:25

Status

carolinaalmeida Accepted 03.05.2025 15:25:51

Number: 2 Author: Subject: Texto insertado Date: 30.03.2025 20:22:31
reflect? show? exhibit?

Status

carolinaalmeida Accepted 03.05.2025 15:26:42

Number: 3 Author: Date: 30.03.2025 20:22:40
, that is, the factors controlling terminal velocity

Status

carolinaalmeida Rejected 03.05.2025 15:28:09

Number: 4 Author: Subject: Texto insertado Date: 30.03.2025 20:22:47

"In the former, interaction between particles and the interstitial gas dominates flow propagation, while in the latter, ..."

I suggest to cite the following paper or something related to fluidization:

Roche, O. (2012). Depositional processes and gas pore pressure in pyroclastic flows: an experimental perspective. *Bulletin of Volcanology*, 74, 1807-1820.

Author: carolinaalmeida Subject: Sticky Note Date: 03.05.2025 15:33:19

Thank you for the suggestion. However, as fluidization processes are not the focus of this study, we believe that including this reference is not necessary for the scope of our analysis.

Number: 5 Author: Subject: Texto insertado Date: 30.03.2025 20:22:55

ve

Status

carolinaalmeida Accepted 03.05.2025 15:34:19

Number: 6 Author: Subject: Texto insertado Date: 30.03.2025 20:23:03
for

Status

carolinaalmeida Rejected 03.05.2025 15:35:30

Number: 7 Author: Date: 30.03.2025 20:23:11

I am not sure I understood the logical step of this sentence starting with "Thus". I suggest to remove it.

Status

carolinaalmeida Accepted 08.05.2025 10:20:31

Author: carolinaalmeida Subject: Sticky Note Date: 03.05.2025 15:41:53

Thank you for the comment. The sentence aims to state that, based on previous experiments, the variability in morphological parameters has been successfully assessed. We have changed "Thus" for "In this sense".

Number: 8 Author: Date: 30.03.2025 20:23:18

Status

carolinaalmeida Rejected 03.05.2025 15:42:26

Number: 9 Author: Date: 30.03.2025 20:23:28

I suggest to mention that the relaxation time has also revealed to be a good framework to model volcanic processes.

Morales Rivera, A. M., Amelung, F., Albino, F., & Gregg, P. M. (2019). Impact of crustal rheology on temperature-dependent viscoelastic models of volcano deformation: Application to Taal Volcano, Philippines. *Journal of Geophysical Research: Solid Earth*, 124(1), 978-994.

de' Michieli, M. & Aravena, A. (2021). Numerical modeling of magma ascent. *Forecasting and Planning for Volcanic Hazards, Risks,*

Status

carolinaalmeida Accepted 03.05.2025 15:43:23

**2. Results**

We have conducted tumbling experiments under **controlled conditions** and quantified
the related abrasion process in terms of the ash fraction and a series of morphological
parameters of particle shape. We have analysed the observed fine particle generation
(section 2.1) and clast shape change (section 2.2) in terms of relaxation times for
shape and roughness evolution.

**2.1 Ash generation**

We present the results of dry tumbling experiments conducted at room temperature
and ambient pressure, expanding an empirical correlation presented by Hornby et al.¹⁷.
We used pumice lapilli from Laacher See (East Eifel, Germany; sourced from fall
deposits of the 13 ka phonolithic Plinian Laacher See eruption near Nickenich,
Germany). Stratigraphically, they belong to the ~~Units~~ Middle (MLST) and Upper (ULST)
Laacher See Tephra ~~and~~ contain 10-15 vol.% (dense rock equivalent) **phenocrysts**³³.

A strong non-linear relationship between tumbling duration (displayed as tumbling
distance – calculated using drum diameter and experiment duration¹⁷) and ash
generated was observed for three experimental conditions of the so-called **Set A (T1,
T2, T3B)** and one for **Set B (T1) experiments**. The ash produced was quantified as the
weight fraction of experimentally generated volcanic ash (based on the starting weight)
and ~~is~~ displayed cumulatively in Fig. 1.

Ash generation was highest at the beginning of the experiments (within the first 1000
meters of tumbling) and decreased non-linearly with increasing ~~Further~~ experimental
duration. This trend can be visualised as a “relaxation” of ash generation efficiency,
dependent on the bulk contact energy per experiment type (see Methods). A sense of
the magnitude of these changes in our experiments is provided by the observation that
amongst all experiments, T3B (i.e., ash removed at each time step and presence of
steel balls) produced the largest amount of ash (up to 48 wt.%), followed by T2 (i.e.,
ash removed at each time step, ca. 36 wt.%), and T1 (i.e., ash returned to the drum
after each time step, ca. 25 wt.%). These results document the prime influence of bulk
contact energy on ash generation. The presence of inter-clast ash led to reduced ash
generation (T1 compared to T2) while the steel balls locally increased point loads that
lead to abrasion (dominant) or break-up (T3 compared to T2).

 Number: 1 Author: Date: 30.03.2025 20:23:43
Which conditions?

Status
carolinaalmeida Accepted 03.05.2025 16:10:36
 Author: carolinaalmeida Subject: Sticky Note Date: 03.05.2025 16:10:33
We added the section where to find the experimental conditions.

 Number: 2 Author: Date: 30.03.2025 20:23:50

Status
carolinaalmeida Accepted 03.05.2025 16:10:53

 Number: 3 Author: Date: 30.03.2025 20:23:58
Which phases?

Status
carolinaalmeida Accepted 03.05.2025 16:23:34
 Author: carolinaalmeida Subject: Sticky Note Date: 03.05.2025 16:23:53
We have added the phases in the text.

 Number: 4 Author: Date: 30.03.2025 20:24:07
units

Status
carolinaalmeida Accepted 03.05.2025 16:12:13

 Number: 5 Author: Date: 30.03.2025 20:24:17

I know it is in the methods, but a small explanation of sets is needed here to facilitate reading. Maybe you can move here the descriptions that currently presented in the next paragraph.

Status
carolinaalmeida Accepted 03.05.2025 16:30:52
 Author: carolinaalmeida Subject: Sticky Note Date: 03.05.2025 16:31:38
We agree and have moved the descriptions to the previous paragraph.

 Number: 6 Author: Subject: Texto insertado Date: 30.03.2025 20:24:23
which

Status
carolinaalmeida Accepted 03.05.2025 16:24:53

 Number: 7 Author: Date: 30.03.2025 20:24:30

Status
carolinaalmeida Accepted 03.05.2025 16:32:31

Figure 1: Ash production during tumbling experiments, plotted as cumulative ash fraction (weight fraction of the starting mass) against the rotational distance for experiments T1, T2 and T3B (*Set A*) and T1 (*Set B*). Result reproducibility is being shown by 2 data sets each (this study) and in accordance with earlier studies¹⁷. *Set B* experiment T1 with coloured clasts reproduces the ash generation trend, proving the subordinate impact of the food colouring on the mechanical properties of the 100 coloured clasts.

2.2 Shape and ~~2~~ Petrophysical analysis

We have extended the analysis and have evaluated 2 ~~3~~ Petrophysical (volume and
 porosity) and 4 morphological (*Axial Ratio* [particle elongation], *Convexity* [fine-scale
 textural roughness], *Form Factor* [overall particle irregularity] and *Solidity*
 [morphological roughness]) parameters that describe the shape evolution during our
 experiments. We have cast the analysis of these experimental results in terms of
 relaxation theory, yielding a novel insight into particle abrasion. Key parameters that
 govern the evolution of volcanic pyroclastic particle shapes are generated, providing a
 useful way for volcanologists to understand transport and sedimentation processes
 (and thereby eruption dynamics) based on thorough analysis of deposits.

More than 2,400 measurements were performed to quantify the shape and
 ~~4~~ Petrophysical properties of pumice lapilli. The morphological evolution of clast shape
 became clear when quantifying shape parameters via 2D image analysis: *Axial Ratio*,
 *Convexity*, *Form Factor*, and *Solidity*. In *Set A*, we analysed 100 randomly selected
 clasts after each time step of tumbling experiments (i.e., 0' [starting material SM], 15',
 30', 45', 60', and 120') (Fig. 2). From the mean values of *Convexity*, *Form Factor* and
 *Solidity* (Box plots, Fig. 2A, 2B and 2C), we observe the fastest evolution in the first 15
 minutes of tumbling.

Page: 4

 Number: 1 Author: Date: 30.03.2025 06:35:26
I suggest to use different line styles as well.

Status

carolinaalmeida Accepted 08.05.2025 10:20:10
 Number: 2 Author: Date: 30.03.2025 22:08:46

Status

carolinaalmeida Accepted 05.05.2025 15:30:35
 Number: 3 Author: Date: 30.03.2025 20:24:44

Status

carolinaalmeida Accepted 05.05.2025 15:30:41
 Number: 4 Author: Date: 30.03.2025 20:24:53

Status

carolinaalmeida Accepted 05.05.2025 15:30:47

Figure 2: Morphological changes in pyroclasts during *Set A* experiments. Particle evolution is shown separately for T1, T2, and T3B as boxplots for (A) *Solidity*, (B) *Convexity* and (C) *Form Factor*. The median is represented by the horizontal line in the box, while the mean is denoted by a cross. Whiskers extend to 1.5 times the interquartile range, with data points beyond this range considered potential outliers.

Set A proved that shape evolution is very rapid.

Regarding overall ash generation (Fig. 1), we observe a relaxation of shape evolution as a function of bulk contact energy per experiment type (Fig. 2). To constrain the observed changes more in detail and rule out any influence of clast heterogeneity when randomly picking 100 clasts, we designed a *Set B* series of experiments (Fig. 3) (for type T1 experiment) and dyed 100 lapilli clasts with food colour, thereby enabling their individual identification and sampling after each experimental increment. Additionally, the shorter time intervals (5', 10', 15', 20', 60') between analytical steps enabled an enhanced resolution of shape evolution with time.

Volume decreased steadily (Fig. 3A), whereas *Axial Ratio* remained constant, confirming the uniform abrasion of tumbling (Fig. 3B). The higher time resolution for *Set B* experiments over *Set A* is

well illustrated by the evolution of the *Convexity* (Fig. 3C), *Form Factor* (Fig. 3D) and *Solidity* (Fig. 3E) parameters. The most substantial changes occurred in the first 5 minutes. After 20 and 60 minutes of tumbling, a very limited number of the starting 100 lapilli had broken as indicated by the increasing number (106 and 120, respectively) of coloured lapilli that could be identified. As the percentage of particles that underwent breakage was relatively small, these breakage events were not frequent enough to generate abrupt parameter changes in the particle population parameters (Fig. 3). The importance of tracking the individual studied particles using the novel techniques employed here (*Set B*) is apparent in our experimental results (Fig. 2 and Fig. 3) since it will allow for modelling particle abrasion.

3. Discussion

Our experimental set-up is related but not identical to the techniques of other studies in volcanology^{16,18,19} and clastic sediment transport^{23,24}. For experiments³⁴ with a comparable horizontal rotational axis, continuous mixing without clast segregation had been observed. Accordingly, for the experimental geometry used here (with slightly

 Number: 1 Author: Date: 30.03.2025 06:44:04
I wonder how representative of natural conditions this is.

 Author: carolinaalmeida Subject: Sticky Note Date: 05.05.2025 15:34:05
Thank you for the comment. We have rephrased the sentence to clarify that we are referring to shape evolution under the tumbling experiments.

 Number: 2 Author: Date: 30.03.2025 06:44:12

Titles above SM are confusing. I suggest to present the labels "Convexity", "Form Factor" and "Solidity" in the y-axis, and a label of SM above the first box in each panel.

Status
carolinaalmeida Accepted 08.05.2025 10:19:59

[revised manuscript text omitted]

The experience gained from the experiments above led us to conclude that a higher
temporal resolution was required to effectively parameterise shape evolution. This was
addressed by designing *Set B* experiments. Here, 100 clasts, dyed with food colour
could be sampled repeatedly after each experimental increment to capture detailed
trends in volume reduction, visualised as changes of *Convexity*, *Form Factor*, *Axial*
*Ratio* and *Solidity* (Fig. 3). The refined temporal trends confirm the strongly nonlinear
~~correlation~~ and indicate that measurable shape changes ensue very rapidly during
transport (Fig. 4b). The observation that the number of coloured starting clasts (n=100)
increased to 106 (three lapilli broke apart) and 120 (ten lapilli broke apart) clasts after
20 and 60 minutes, respectively, underscores the presence but not the dominance of
clast breakage as source of ash particles during tumbling. The breakage of these clasts
was not significant enough to cause discontinuities or abrupt changes in the trends
(Fig. 3), indicating that particle abrasion is the dominant mechanism driving shape
evolution during the tumbling experiments.

 Number: 1 Author: Date: 30.03.2025 20:26:15

Status

carolinaalmeida Accepted 05.05.2025 15:36:55

 Number: 2 Author: Subject: Texto insertado Date: 30.03.2025 20:30:13

shape evolution dynamics of pyroclasts

Status

carolinaalmeida Accepted 05.05.2025 15:38:05

Granular materials exhibit variability in both their morphological properties, such as
 grain shapes, size distributions, and surface characteristics; and their microstructural
 arrangements, including the organisation of grains and their contact points³⁷. In our
 tumbling experiments, we do not address the microstructural arrangements due to the
 complexity and variability of how grains organise and interact at a small scale.
 Interactions between particles of dry, rough, and granular materials have been
 experimentally defined as being mainly due to friction and collision³⁸, and
 fragmentation and abrasion are the main processes that govern disruption of the
 frictional materials³⁹. In collisional interactions with a significant overlap of clast cross-
 sectional area (>50%), particle break-up is promoted, whereas in the tangential case

with overlap (<10%), impact energy is predominantly translated into rotational movement, yielding abrasion. Based on the here observed propensity of porous pyroclasts to shape change, we conclude from the angular lapilli clasts observed in fall deposits of explosive eruptions indicate that collisional (promoting disruption) and/or tangential (promoting abrasion) interactions with other pyroclasts or the conduit walls are of subordinate importance there.

[revised manuscript text omitted]

Number: 1 Author: Date: 31.03.2025 05:37:58
Are we sure about this? I mean, there are other factor such as temperature.

Author: carolinaalmeida Subject: Sticky Note Date: 05.08.2025 15:02:31
Thank you for the comment. Yes, even considering temperature and other factors, particle velocities (and thus kinetic energies) in our tumbling experiments are orders of magnitude lower than in natural currents.

Number: 2 Author: Date: 30.03.2025 21:02:08
Ok, but at some point the eruption is nearly steady-state.

Author: carolinaalmeida Subject: Sticky Note Date: 05.08.2025 15:08:06
However, even in near steady-state phases, since no eruption is actually steady-state, particle velocity remains lower than at peak discharge due to declining overpressure.

Number: 3 Author: Date: 30.03.2025 21:02:41

Status
carolinaalmeida Accepted 08.05.2025 10:01:35

Number: 4 Author: Date: 30.03.2025 21:02:34

Status
carolinaalmeida Accepted 08.05.2025 10:01:54

Number: 5 Author: Date: 30.03.2025 21:05:09

Status
carolinaalmeida Accepted 08.05.2025 10:03:00

Number: 6 Author: Subject: Texto insertado Date: 30.03.2025 21:05:02

Status
carolinaalmeida Accepted 08.05.2025 10:03:06

Number: 7 Author: Subject: Texto insertado Date: 30.03.2025 21:04:55

Status
carolinaalmeida Accepted 08.05.2025 10:04:00

Number: 8 Author: Subject: Texto insertado Date: 30.03.2025 21:09:34

each presenting

Status
carolinaalmeida Accepted 08.05.2025 10:04:27

When a material undergoes a certain change, any feature of physical property at a
given time $p(t)$ evolves with aging time from its initial value p_0 to its final value p_∞ ,
such that a normalised relaxation function $\phi(t)$ is the following:

$$329 \quad \phi(t) = \frac{p(t) - p_\infty}{p_0 - p_\infty}. \quad [\text{eq. 1}]$$

We employ the Kohlrausch-Williams-Watts (KWW) equation (eq. 2) that describes the
temporal function decay of a given property p ^{29,30}:

$$332 \quad \phi(t) = \exp \left[- \left(\frac{t}{\tau_k} \right)^\beta \right], \quad [\text{eq. 2}]$$

where τ_k is the characteristic relaxation time, and β describes the breadth of the
relaxation time distribution, serving as the non-exponentiality parameter. In case β is
unity, it corresponds to a simple case of exponential decay, representing a single
relaxation time, i.e., the process happens all over the material at the same rate. For 0
$< \beta < 1$, it indicates a stretched exponential decay, meaning a distribution of relaxation
338 times, where β close to unity represents a narrow distribution, while β close to zero
represents a broad range of relaxation times across the material. A combination of
equations eq.1 and eq.2 gives rise to:

$$341 \quad p(t) = p_\infty + (p_0 - p_\infty) \exp \left[- \left(\frac{t}{\tau_k} \right)^\beta \right] \quad [\text{eq. 3}]$$

We used the mean values of *Convexity*, *Form Factor*, and *Solidity* for each time
increment (0, 5, 10, 15, 20, and 60 minutes) of experiment T1 (*Set B*) as the studied
property p (Fig. 5). Time increment 0 refers to the starting material (SM). The relaxation
timescale for *Convexity* is 5.1 minutes (Fig. 5A), *Form Factor* 8.7 minutes (Fig. 5B) and
*Solidity* 12.5 minutes (Fig. 5C), respectively, implying that finer-scale adjustments
occur on shorter timescales than changes to the overall shape and structural
roughness of the particles.

By analysing eq. 3, τ_k refers to the time required for particles relaxation 63.2 % of its
path to equilibrium if $\beta = 1$. However, for this studied case, β is not unity (see Fig. 5).
Therefore, for our studied case instead of 63.2 % we obtained 42 %, 50 % and 43 %
of its path to equilibrium for *Convexity*, *Form Factor*, and *Solidity*, respectively. As
mentioned, β reflects the distribution of relaxation times, which in glass science is
linked to the varying structures on atomic to nanoscale levels⁵⁰⁻⁵², therefore these
different regions undergo relaxation following different rates. By analogy with the
current system, we suggest that the β values (cf. Fig. 5) illustrate how different particles
undergo shape relaxation and the close it is to unity means that particles change shape
in the same rate (narrow distribution of shape relaxation time), while for β going
towards zero, means a broad distribution of shape relaxation times. The analysis of
this parameter can be useful tool to evaluate how heterogeneous our set of particles
is. Interestingly, comparing the experiments with and without particle tracking (using
food colouring) yielded different results, characterising our system with β values not

 Number: 1 Author: Subject: Texto insertado Date: 30.03.2025 21:15:02
,equilibrium

Status

carolinaalmeida Rejected 08.05.2025 10:07:37

 Number: 2 Author: Date: 30.03.2025 21:21:01

Please correct the subscript here.

Status

carolinaalmeida Accepted 08.05.2025 10:07:52

Figure 5: Temporal evolution of particle shape parameters during tumbling experiment type T1. The plots depict the variation of (A) Solidity, S, (B) Convexity, C, and (C) Form Factor, FF, over time t [min]. The solid black lines represent the best fit to the experimental data (red dots) using the relaxation model described by equation [1], and the initial (SM) and final values are provided within each plot.

currents. Eventually, this will serve to educate scientists advising the population on and
 around volcanoes and ultimately reduce the lethality of such terrible events.

4. Methods

near unity, as different particles change shape in different rates (cf. Fig. 2 and Fig. 3). These variations might be related to different textures of the starting materials, such as crystal and bubble content and shape.

The parameterisation of the evolution of pyroclast morphology presented here enhances our physical understanding of transport of pyroclastic materials, in detail the efficiency of particle-particle interaction and related ash generation. We believe this to be of great significance as collisional processes take place in a wide variety of volcanic settings. Pyroclasts formed by brittle fragmentation and preserved as angular fragments in fall deposits serve as proxy for natural processes and serve as reference point against which pyroclasts collected from PDC deposits may be compared. We suggest the use of relaxation theory to describe the shape evolution of particle in terms of *Convexity, Form Factor and Solidity*. Thorough quantitative analysis of clasts from primary volcanic deposits directly in the field will avoid any post-sampling size and shape alteration. We propose that this will allow for upscaling the empirical laws from laboratory experiments and contribute to enhanced understanding of transportation processes, in particular those affecting the mobility of density

 Number: 1 Author: Subject: Texto insertado Date: 30.03.2025 21:45:58
improves

Status

carolinaalmeida Accepted 08.05.2025 10:09:12

 Number: 2 Author: Subject: Texto insertado Date: 30.03.2025 21:22:01
while

Status

carolinaalmeida Accepted 08.05.2025 10:12:27

 Number: 3 Author: Date: 30.03.2025 22:09:18
pyroclastic

Status

carolinaalmeida Accepted 08.05.2025 10:13:16

 Number: 4 Author: Date: 30.03.2025 21:47:48

Status

carolinaalmeida Accepted 08.05.2025 10:13:21

Rotary tumbling experiments that mimic particle interactions during PDCs or granular
 flows have been used to study the abrasion of pumice lapilli. The particle interactions,
 albeit likely of lesser intensity than possible in natural transport, are caused by forced
 tumbling of the experimental charge. Previous studies¹⁷⁻¹⁹ have empirically constrained
 the relationship between tumbling conditions and ash generation. To increase our
 mechanistic understanding of pyroclast size and shape evolution during PDC
 transport, we conducted further tumbling experiments as defined below. We extended
 the analysis of the experimental cargo and have evaluated two ~~1~~etrophysical (volume
 and porosity) and four morphological (*Axial Ratio* [particle elongation], *Convexity* [fine-
 scale textural roughness], *Form Factor* [overall particle irregularity] and *Solidity*
 [morphological roughness]) parameters that describe the shape evolution during our
 experiments.

4.1 Tumbling experiments and ash generation

The tumbling experiments were carried out at ambient conditions in an industrial
 cement mixer (see Hornby et al.¹⁷ for detailed description). The starting material (SM)
 was sourced from fall deposits of the 13 ka phonolithic Plinian Laacher See eruption
 near Nickenich, Germany. Each experiment started with 2 kg of pumice lapilli
 (n~3.000) with a porosity range of 45-93% -mean porosity 80%- and particle size of 2-

Set	Experiment type	Description	Duration	Target clasts
A	T1	Ash and lapilli returned to drum in each step for further tumbling	15', 30', 45', 60', 120'	random (n=100)
	T2	Only lapilli returned to drum, ash stored separately in each step		
	T3B	T2 + 2 steel balls (220g each)		
B	T1	Ash and lapilli returned to drum in each step for further tumbling	5', 10', 15', 20', 60'	coloured (n=100)

Table 1: Description of the experimental conditions of two sets and three types of tumbling experiments. Collisional energy was varied by removing/keeping ash from/in the drum after each increment or adding steel balls. Clasts have been investigated after five time-steps (representing theoretical tumbling distance). In *Set A*, random clasts were selected for analysis, in *Set B*, the same batch of 100 clasts was investigated after each increment.

5 cm. In *Set A*, three types of experiments were performed to map out abrasion
 efficiency as a function of collisional energy. In T1 experiments, lapilli and ash
 generated were put back into the drum after each time increment. In T2, only the
 remaining lapilli fraction was put back into the drum, the ash was sampled and stored.
 In T3B, we added two steel balls to mimic the frequent presence of higher-density (than
 pumice) clasts during natural transport. Only the lapilli fraction was put back into the
 drum after each time increment. Ash generation was quantified by dry sieving at 2 mm
 after 15, 30, 45, 60 and 120 minutes, respectively (Table 1).

For T1, we performed a second set of experiments (*Set B*), in which 100 lapilli from
 within the 2 kg starting material had been coloured with food colour prior to tumbling.

 Number: 1 Author: Date: 30.03.2025 22:09:29

Status

carolinaalmeida Accepted 08.05.2025 10:13:29

 Number: 2 Author: Date: 30.03.2025 21:54:30

Is there a reason why some numbers are in bold letters?

Status

carolinaalmeida Accepted 08.05.2025 10:19:34

 Author: carolinaalmeida Subject: Sticky Note Date: 08.05.2025 10:19:09

Thank you for the comment. We have now clarified in the manuscript that the bold formatting of the duration numbers indicates the time steps that are repeated in both experimental sets.

This treatment did not affect the mechanical response as density and weight remained
 effectively unchanged. Ash generation was quantified by dry sieving at 2 mm after 5,
 10, 15, 20 and 60 minutes, respectively.

4.2 Petrophysical properties and shape analysis

In addition to obtaining the weight fraction of ash generated per experimental
 increment, we also quantified clast volume (Fig. 6a), weight and morphology (Fig. 6b).
 Clast volumes were calculated using the manual method⁵³, measuring the length of
 three orthogonal axes with a calliper and calculating the volume assuming a three-axial
 ellipsoid geometry. High-precision weight analysis (10^{-4} g) allowed for calculating the
 porosity of each clast. For shape analysis, we used the projected area of the clasts
 from digital photographs, processed the images on Photoshop (clast contours
 delimitation and binarization) and calculated shape parameters on *ImageJ*. We
 increased the resolution compared to earlier studies¹⁷ by analysing macro images
 taken of each clast individually (and not in sets of $n=100$). We selected *Axial Ratio*,
 *Convexity*, *Form Factor* and *Solidity*⁵⁴ and used a adapted shape macro⁵⁵. In *Set A*,
 100 randomly selected clasts were taken from the lapilli fraction for analysis and
 afterwards put back to the lapilli cargo in the drum for the subsequent tumbling. In *Set B*,
 clasts coloured with food colour could be picked after each experimental increment
 for analysis and afterwards put back to the lapilli cargo in the drum for the subsequent
 tumbling.

Figure 6: Workflow to constrain the petrophysical properties, volume (a) and shape (b). a) Calliper measurement of three orthogonal axes (A, B, and C) and volume calculation. b) Illustration of the parametrization measurements (*Axial Ratio*, *Convexity*, *Form Factor* and *Solidity*) on a binary image of pumice clast.

Acknowledgements

CF and UK acknowledge support by grant KU2689/7-1 from the Deutsche
 Forschungsgemeinschaft. LP is grateful for the support of the Alexander von Humboldt
 Foundation. DBD, LP, & CF acknowledge the support of ERC 2018 ADV Grant 834225
 (EAVESDROP) to DBD.

References

- 1 Dingwell, D. B. & Webb, S. L. Structural relaxation in silicate melts and non-Newtonian
 melt rheology in geologic processes. *Physics and Chemistry of Minerals* **16** (1989).
 <https://doi.org/10.1007/bf00197020>

 Number: 1 Author: Date: 30.03.2025 22:09:40

Status

carolinaalmeida Accepted 08.05.2025 10:19:25

 Number: 2 Author: Date: 30.03.2025 22:09:47

Status

carolinaalmeida Accepted 08.05.2025 10:19:29

- Dingwell, D. B. Volcanic Dilemma--Flow or Blow? *Science* **273**, 1054-1055 (1996).
<https://doi.org/10.1126/science.273.5278.1054>
- Kueppers, U., Perugini, D. & Dingwell, D. B. "Explosive energy" during volcanic
eruptions from fractal analysis of pyroclasts. *Earth and Planetary Science Letters* **248**,
800-807 (2006). <https://doi.org/10.1016/j.epsl.2006.06.033>
- Spieler, O. *et al.* The fragmentation threshold of pyroclastic rocks. *Earth and Planetary*
*Science Letters* **226**, 139-148 (2004). <https://doi.org/10.1016/j.epsl.2004.07.016>
- Carey, S. N. & Sigurdsson, H. Influence of particle aggregation on deposition of distal
tephra from the M_{Ay} 18, 1980, eruption of Mount St. Helens volcano. *Journal of*
*Geophysical Research: Solid Earth* **87**, 7061-7072 (1982).
<https://doi.org/10.1029/JB087iB08p07061>
- Pyle, D. M. The thickness, volume and grainsize of tephra fall deposits. *Bulletin of*
*Volcanology* **51**, 1-15 (1989). <https://doi.org/10.1007/bf01086757>
- Sparks, R. S. J., Bursik, M. I., Ablay, G. J., Thomas, R. M. E. & Carey, S. N. Sedimentation
of tephra by volcanic plumes. Part 2: controls on thickness and grain-size variations of
tephra fall deposits. *Bulletin of Volcanology* **54**, 685-695 (1992).
<https://doi.org/10.1007/bf00430779>
- Wilson, L. & Huang, T. C. The influence of shape on the atmospheric settling velocity
of volcanic ash particles. *Earth and Planetary Science Letters* **44**, 311-324 (1979).
[https://doi.org/10.1016/0012-821x\(79\)90179-1](https://doi.org/10.1016/0012-821x(79)90179-1)
- Dufek, J. & Bergantz, G. W. Suspended load and bed-load transport of particle-laden
gravity currents: the role of particle–bed interaction. *Theoretical and Computational*
*Fluid Dynamics* **21**, 119-145 (2007). <https://doi.org/10.1007/s00162-007-0041-6>
- Jones, T. J., Shetty, A., Chalk, C., Dufek, J. & Gonnermann, H. M. Identifying rheological
regimes within pyroclastic density currents. *Nature Communications* **15**, 4401 (2024).
<https://doi.org/10.1038/s41467-024-48612-7>
- Taddeucci, J. & Palladino, D. Particle size-density relationships in pyroclastic deposits:
inferences for emplacement processes. *Bulletin of Volcanology* **64**, 273-284 (2002).
<https://doi.org/10.1007/s00445-002-0205-6>
- Valentine, G. A. Stratified flow in pyroclastic surges. *Bulletin of Volcanology* **49**, 616-
(1987). <https://doi.org/10.1007/bf01079967>
- Buckland, H. M., Eychenne, J., Rust, A. C. & Cashman, K. V. Relating the physical
properties of volcanic rocks to the characteristics of ash generated by experimental
abrasion. *Journal of Volcanology and Geothermal Research* **349**, 335-350 (2018).
<https://doi.org/10.1016/j.jvolgeores.2017.11.017>
- Dufek, J. & Manga, M. In situ production of ash in pyroclastic flows. *Journal of*
*Geophysical Research: Solid Earth* **113** (2008). <https://doi.org/10.1029/2007jb005555>
- Walker, G. P. L. Generation and dispersal of fine ash and dust by volcanic eruptions.
*Journal of Volcanology and Geothermal Research* **11**, 81-92 (1981).
[https://doi.org/10.1016/0377-0273\(81\)90077-9](https://doi.org/10.1016/0377-0273(81)90077-9)
- Cagnoli, B. & Manga, M. Granular mass flows and Coulomb's friction in shear cell
experiments: Implications for geophysical flows. *Journal of Geophysical Research:*
*Earth Surface* **109** (2004). <https://doi.org/10.1029/2004jf000177>
- Hornby, A., Kueppers, U., Maurer, B., Poetsch, C. & Dingwell, D. Experimental
constraints on volcanic ash generation and clast morphometrics in pyroclastic density
currents and granular flows. *Volcanica* **3**, 263-283 (2020).
<https://doi.org/10.30909/vol.03.02.263283>

- 18 Kueppers, U., Putz, C., Spieler, O. & Dingwell, D. B. Abrasion in pyroclastic density
currents: Insights from tumbling experiments. *Physics and Chemistry of the Earth, Parts*
*A/B/C* **45-46**, 33-39 (2012). <https://doi.org/10.1016/j.pce.2011.09.002>
- 19 Manga, M., Patel, A. & Dufek, J. Rounding of pumice clasts during transport: field
measurements and laboratory studies. *Bulletin of Volcanology* **73**, 321-333 (2011).
<https://doi.org/10.1007/s00445-010-0411-6>
- 20 Mueller, S. B., Lane, S. J. & Kueppers, U. Lab-scale ash production by abrasion and
collision experiments of porous volcanic samples. *Journal of Volcanology and*
*Geothermal Research* **302**, 163-172 (2015).
<https://doi.org/10.1016/j.jvolgeores.2015.07.013>
- 21 Schwarzkopf, L. M., Spieler, O., Scheu, B. & Dingwell, D. B. Fall-experiments on Merapi
basaltic andesite and constraints on the generation of pyroclastic surges. *eEarth* **2**, 1-
5 (2007). <https://doi.org/10.5194/ee-2-1-2007>
- Caballero, L., Sarocchi, D., Borselli, L. & Cárdenas, A. I. Particle interaction inside debris
flows: Evidence through experimental data and quantitative clast shape analysis.
*Journal of Volcanology and Geothermal Research* **231-232**, 12-23 (2012).
<https://doi.org/10.1016/j.jvolgeores.2012.04.007>
- Yao, T., Yang, H., Lourenço, S. D. N., Baudet, B. A. & Kwok, F. C. Y. Multi-scale particle
morphology evolution in rotating drum tests: Role of particle shape and pore fluid.
*Engineering Geology* **303** (2022). <https://doi.org/10.1016/j.enggeo.2022.106669>
- Lewin, J. & Brewer, P. A. Laboratory simulation of clast abrasion. *Earth Surface*
*Processes and Landforms* **27**, 145-164 (2002). <https://doi.org/10.1002/esp.306>
- Deal, E. *et al.* Grain shape effects in bed load sediment transport. *Nature* **613**, 298-302
(2023). <https://doi.org/10.1038/s41586-022-05564-6>
- Kragelsky, I. V. (eds I.V. Kragelsky, M.N. Dobyichin, & V.S. Kombalov) 297–316
(Pergamon, 1982).
- Campbell, I. A. & Giovannella, C. Relaxation in Complex Systems and Related Topics.
*Springer* (1990). <https://doi.org/10.1007/978-1-4899-2136-9>
- Jaeger, H. M. Celebrating Soft Matter's 10th Anniversary: Toward jamming by design.
*Soft Matter* **11**, 12-27 (2015). <https://doi.org/10.1039/c4sm01923g>
- Kohlrausch, R. Theorie des elektrischen Rückstandes in der Leidener Flasche. *Annalen*
*der Physik* **167**, 179-214 (1854). <https://doi.org/10.1002/andp.18541670203>
- Williams, G. & Watts, D. C. Non-symmetrical dielectric relaxation behaviour arising
from a simple empirical decay function. *Transactions of the Faraday Society* **66** (1970).
<https://doi.org/10.1039/tf9706600080>
- Lancelotti, R. F., Zanotto, E. D. & Sen, S. Kinetics of physical aging of a silicate glass
following temperature up- and down-jumps. *The Journal of Chemical Physics* **160**
(2024). <https://doi.org/10.1063/5.0185538>
- Lancelotti, R. F., Pereira, L., Hess, K.-U., Dingwell, D. B. & Zanotto, E. D. Flash-DSC
provides valuable insights into glass relaxation and crystallization. *Journal of Non-*
*Crystalline Solids* **646** (2024). <https://doi.org/10.1016/j.jnoncrsol.2024.123242>
- Schmincke, H.-U., Park, C. & Harms, E. Evolution and environmental impacts of the
eruption of Laacher See Volcano (Germany) 12,900 a BP. *Quaternary International* **61**,
61-72 (1999). [https://doi.org/10.1016/s1040-6182\(99\)00017-8](https://doi.org/10.1016/s1040-6182(99)00017-8)
- Cleary, P. W. DEM simulation of industrial particle flows: case studies of dragline
excavators, mixing in tumblers and centrifugal mills. *Powder Technology* **109**, 83-104
(2000). [https://doi.org/10.1016/s0032-5910\(99\)00229-6](https://doi.org/10.1016/s0032-5910(99)00229-6)

- Freundt, A. & Schmincke, H. U. Abrasion in pyroclastic flows. *Geologische Rundschau*
**81**, 383-389 (1992). <https://doi.org/10.1007/bf01828605>
- Breard, E. C. P. *et al.* The fragmentation-induced fluidisation of pyroclastic density
currents. *Nature Communications* **14** (2023). [https://doi.org/10.1038/s41467-023-](https://doi.org/10.1038/s41467-023-37867-1)
[37867-1](https://doi.org/10.1038/s41467-023-37867-1)
- Radjai, F., Roux, J.-N. & Daouadji, A. Modeling Granular Materials: Century-Long
Research across Scales. *Journal of Engineering Mechanics* **143** (2017).
[https://doi.org/10.1061/\(asce\)em.1943-7889.0001196](https://doi.org/10.1061/(asce)em.1943-7889.0001196)
- Yu, F. *et al.* Particle breakage of sand subjected to friction and collision in drum tests.
*Journal of Rock Mechanics and Geotechnical Engineering* **13**, 390-400 (2021).
<https://doi.org/10.1016/j.jrmge.2020.08.004>
- Kirchner, N. P. Thermodynamically consistent modelling of abrasive granular materials.
I Non-equilibrium theory. *Proceedings of the Royal Society of London. Series A:*
*Mathematical, Physical and Engineering Sciences* **458**, 2153-2176 (2002).
<https://doi.org/10.1098/rspa.2002.0963>
- Cigala, V. *et al.* The dynamics of volcanic jets: Temporal evolution of particles exit
velocity from shock-tube experiments. *Journal of Geophysical Research: Solid Earth*
**122**, 6031-6045 (2017). <https://doi.org/10.1002/2017jb014149>
- Daniel, R. C., Poloski, A. P. & Eduardo Sáez, A. A continuum constitutive model for
cohesionless granular flows. *Chemical Engineering Science* **62**, 1343-1350 (2007).
<https://doi.org/10.1016/j.ces.2006.11.035>
- Fang, C., Wang, Y. & Hutter, K. A thermo-mechanical continuum theory with internal
length for cohesionless granular materials. *Continuum Mechanics and*
*Thermodynamics* **17**, 545-576 (2006). <https://doi.org/10.1007/s00161-006-0007-8>
- Fang, C., Wang, Y. & Hutter, K. Shearing flows of a dry granular material—hypoplastic
constitutive theory and numerical simulations. *International Journal for Numerical and*
*Analytical Methods in Geomechanics* **30**, 1409-1437 (2006).
<https://doi.org/10.1002/nag.525>
- Fang, C. & Wu, W. On the weak turbulent motions of an isothermal dry granular dense
flow with incompressible grains: part I. Equilibrium turbulent closure models. *Acta*
*Geotechnica* **9**, 725-737 (2014). <https://doi.org/10.1007/s11440-014-0313-4>
- Göncü, F. & Luding, S. Effect of particle friction and polydispersity on the macroscopic
stress-strain relations of granular materials. *Acta Geotechnica* **8**, 629-643 (2013).
<https://doi.org/10.1007/s11440-013-0258-z>
- Volfson, D., Tsimring, L. S. & Aranson, I. S. Partially fluidized shear granular flows:
continuum theory and molecular dynamics simulations. *Phys Rev E Stat Nonlin Soft*
*Matter Phys* **68**, 021301 (2003). <https://doi.org/10.1103/PhysRevE.68.021301>
- Bagdassarov, N. S. & Dingwell, D. B. A rheological investigation of vesicular rhyolite.
*Journal of Volcanology and Geothermal Research* **50**, 307-322 (1992).
[https://doi.org/10.1016/0377-0273\(92\)90099-y](https://doi.org/10.1016/0377-0273(92)90099-y)
- Bagdassarov, N. S. & Dingwell, D. B. Frequency dependent rheology of vesicular
rhyolite. *Journal of Geophysical Research: Solid Earth* **98**, 6477-6487 (1993).
<https://doi.org/https://doi.org/10.1029/92JB02690>
- Bagdassarov, N. S. & Dingwell, D. B. Deformation of foamed rhyolites under internal
and external stresses: an experimental investigation. *Bulletin of Volcanology* **55**, 147-
154 (1993). <https://doi.org/10.1007/bf00301512>

- Cormier, L., Galois, L., Lelong, G. & Calas, G. From nanoscale heterogeneities to
nanolites: cation clustering in glasses. *Comptes Rendus. Physique* **24**, 199-214 (2023).
<https://doi.org/10.5802/crphys.150>
- Greaves, G. N. EXAFS and the structure of glass. *Journal of Non-Crystalline Solids* **71**,
203-217 (1985). [https://doi.org/10.1016/0022-3093\(85\)90289-3](https://doi.org/10.1016/0022-3093(85)90289-3)
- Le Losq, C. *et al.* Percolation channels: a universal idea to describe the atomic structure
and dynamics of glasses and melts. *Scientific Reports* **7** (2017).
<https://doi.org/10.1038/s41598-017-16741-3>
- Pisello, A. *et al.* The porosity of felsic pyroclasts: laboratory validation of field-based
approaches. *Bulletin of Volcanology* **85**, 69 (2023). [https://doi.org/10.1007/s00445-](https://doi.org/10.1007/s00445-023-01679-4)
[023-01679-4](https://doi.org/10.1007/s00445-023-01679-4)
- Liu, E. J., Cashman, K. V. & Rust, A. C. Optimising shape analysis to quantify volcanic
ash morphology. *GeoResJ* **8**, 14-30 (2015). <https://doi.org/10.1016/j.grj.2015.09.001>
- Figueiredo, C. A., Bongiolo, E. M., Jutzeler, M., da Fonseca Martins Gomes, O. &
Neumann, R. Alkaline pyroclast morphology informs on fragmentation mechanisms,
Trindade Island, Brazil. *Journal of Volcanology and Geothermal Research* **428** (2022).
<https://doi.org/10.1016/j.jvolgeores.2022.107575>

**Author contributions.** CF and UK conceptualized the study. CF, UK, LD and SE
performed the experiments and analysed the data. LP modelled the experimental data.
CF and UK wrote the manuscript. DBD provided the funding. All authors discussed the
findings, developed the analysis and generated the manuscript.

Reviewer 1 comments:

The use of an experimental setting by steps should be mentioned previously (Line 23).

We thank the reviewer for the comment but we disagree with the suggestion. The explanation of the experimental setting as using steps is given in the methodology.

It is difficult to understand if this is a short or long period if the total duration of experiments is not presented previously (Line 26).

Thank you for the comment, we have added the information that 15 minutes is equivalent of the first experimental step, so it is possible for the reader to understand that we are talking about a very short period.

transport dynamics? (Line 43)

We thank the reviewer for the comment, and we agree with his suggestion. Fall and flow are the transport dynamics and not the deposit types. We have changed accordingly.

Present (Line 50)

We thank the reviewer for the suggestion and have modified accordingly.

As a consequence of this process, in medial (Line 54)

We appreciate the reviewer contribution and have added to the manuscript.

transport regimes for (Line 57)

We thank the reviewer for their input, which has been added to the revised manuscript.

Strikethrough text (Line 59)

We thank the reviewer for the suggestion; however, we believe that the term "porous" is important to keep, as it highlights that the study specifically refers to pumice.

and significantly altering grain size distribution (Line 60)

We thank the reviewer for the comment and have rephrased the sentence to indicate that pyroclasts not only become more rounded but also smaller.

What do you mean with "the experimental condition"? Please reformulate (Line 63)

We thank the reviewer and have changed to experimental settings to clarify.

These? (Line 67)

We agree and have changed accordingly.

step? (Line 122)

We recognize the mistake and have changed 'stop' to 'step'.

The plots order is not consistent with the caption (Figure 2 caption)

We thank the reviewer for noticing this inconsistency. We have corrected the order of the plots in the Figure 2 caption.

I suggest to use a regular time step in the x-axis, or including a mark to show a skip between 20' a 60' (Figure 3)

We thank the reviewer for the suggestion, and we have included a mark to show a skip between 20' and 60' on Figure 3 and 60' and 120' on Figure 2.

I suggest to highlight the innovative aspects in this part (Line 180)

We thank the reviewer for the suggestion, however we do not think that it is necessary to highlight the innovative aspects in this part. We have already explored it on lines (218-245).

Regime (Line 191)

We agree and have added the information accordingly.

I suggest to extent the analysis of the effect of fluidization in increasing runout distance (Line 190-218)

We thank the reviewer for the suggestion. We believe that the effect of fluidization on runout distance is already sufficiently discussed in the text, and the relevant reference is included in this section.

I suggest to slightly expand the reasons for these dynamic variations in nature (Line 232-233)

We thank the reviewer for the suggestion and we have added the possible dynamic variations in nature.

decrease? (Line 240)

We thank the reviewer for pointing this out and we have deleted the word 'reduces' since it has already been used previously and can be used to indicate 'intensity of clast interactions' and 'particle abrasion'.

Strikethrough text (Line 290)

We agree with the reviewer and have deleted the word 'Here'.

I think this should be included before to increase clarity (In L326, before talking about PDCs) (Line 334-339)

We appreciate the reviewer's consideration. We have moved the indicated lines to the beginning of the paragraph in order to better introduce the topic before elaborating on it.

,supporting (Line 342)

We thank the reviewer's comment and have changed accordingly.

Strikethrough text (Line 365)

We thank the reviewer's comment and have altered the words 'dependent of' to 'controlled by' to increase clarity.

Strikethrough text (Line 371)

We thank the reviewer for the suggestion; however we think that keeping 'application of' is important for maintaining the logical flow of the argument.

Strikethrough text (Line 394)

We acknowledge the reviewer's comment, and this has been modified in the revised version.

Strikethrough text (Line 400)

We have considered the reviewer's suggestion and have incorporated this modification into the revised manuscript.

a (Line 422)

We have included it in the revised text.

This part is a little bit confusing (Lines 421-427)

We thank the reviewer for pointing this out and have rephrased the sentence for better understanding.

The subplots order is not consistent with the caption (Figure 5)

We acknowledge the reviewer's comment, and we have changed it accordingly.

Strikethrough text (Line 474)

We thank the reviewer for the suggestion and have modified it in the revised text.

Strikethrough text (Line 479)

We have implemented the reviewer's suggestion in the revised text.